# Tree of Preferences for Diversified Recommendation

**Hanyang Yuan**[1][†][‡], **Ning Tang**[2][†], **Tongya Zheng**[3][*][§], **Jiarong Xu**[2][*], **Xintong Hu**[1]
**Renhong Huang**[1], **Shunyu Liu**[4], **Jiacong Hu**[1], **Jiawei Chen**[1], **Mingli Song**[1]
[1]Zhejiang University, [2]Fudan University
[3]Hangzhou City University, [4]Nanyang Technological University
{yuanhanyang, 320010383, renh2}@zju.edu.cn
{jiaconghu, sleepyhunt, brooksong}@zju.edu.cn
ningtang24@m.fudan.edu.cn, jiarongxu@fudan.edu.cn
doujiang_zheng@163.com, shunyu.liu@ntu.edu.sg

## Abstract

Diversified recommendation has attracted increasing attention from both researchers and practitioners, which can effectively address the homogeneity of recommended items. Existing approaches predominantly aim to infer the diversity of user preferences from observed user feedback. Nonetheless, due to inherent data biases, the observed data may not fully reflect user interests, where *underexplored* preferences can be overwhelmed or remain unmanifested. Failing to capture these preferences can lead to suboptimal diversity in recommendations. To fill this gap, this work aims to study diversified recommendation from a data-bias perspective. Inspired by the outstanding performance of large language models (LLMs) in zero-shot inference leveraging world knowledge, we propose a novel approach that utilizes LLMs' expertise to uncover underexplored user preferences from observed behavior, ultimately providing diverse and relevant recommendations. To achieve this, we first introduce Tree of Preferences (ToP), an innovative structure constructed to model user preferences from coarse to fine. ToP enables LLMs to systematically reason over the user's rationale behind their behavior, thereby uncovering their underexplored preferences. To guide diversified recommendations using uncovered preferences, we adopt a data-centric approach, identifying candidate items that match user preferences and generating synthetic interactions that reflect underexplored preferences. These interactions are integrated to train a general recommender for diversification. Moreover, we scale up overall efficiency by dynamically selecting influential users during optimization. Extensive evaluations of both diversity and relevance show that our approach outperforms existing methods in most cases and achieves near-optimal performance in others, with reasonable inference latency.

## 1 Introduction

Recommender systems have gained significant value in recent years, powering diverse applications such as social media [1, 2], e-commerce [3, 4], and video streaming platforms [5, 6]. To improve recommendation performance, numerous models have emerged, which are essentially built upon interactions between users and items [4, 7–10], such as shares, likes in social media, ratings in

---

[†]Equal contribution.
[*]Corresponding authors.
[‡]State Key Laboratory of Blockchain and Security, Zhejiang University.
[§]Zhejiang Provincial Engineering Research Center for Real-Time SmartTech in Urban Security Governance, School of Computer and Computing Science, Hangzhou City University.

39th Conference on Neural Information Processing Systems (NeurIPS 2025).

e-commerce, or viewing time in video streaming[11–14]. While these systems strive to optimize alignment with user behavior data, it is increasingly recognized that user feedback, typically based on observed data, carries inherent biases, which can have various impacts on recommendation systems [7, 10, 15]. Among these, recommendation *diversity* is an aspect crucial to user satisfaction.

Recommendation diversity represents the dissimilarity of items recommended to an individual user [16]. Numerous studies have shown that higher diversity can provide users with freshness, meet their diverse interests, and lead to higher engagement [1, 17]. Recent works have explored various methods to enhance recommendation diversity. Some focus on the reranking phase, leveraging user-behavior models [17] or graph clustering [1] to capture user interests. Others focus on the matching stage, including multi-vector representations [7], separate matching of categories and items [9], or box embedding [6]. Another set of approaches comes from data perspective, such as node-copying for diverse sample graphs [8], knowledge distillation from diversified teachers [18], or rebalanced neighbor sampling [16]. Although these methods make considerable progress, they rely solely on existing observed user-item interactions, without supplementary knowledge. However, due to inherent biases in the data itself (*e.g.*, exposure bias, selection bias [10]), it may not be sufficient to comprehensively infer user preferences. Consequently, to improve recommendation diversity, these methods can carry a higher risk of irrelevant suggestions, leading to a degradation in recommendation relevance, which contradicts the fundamental goal of accuracy for recommender systems.

Ideally, recommender systems are expected to capture comprehensive aspects of user interests based on observed user behavior, thereby providing a diversified set of recommendations. Nevertheless, due to biases, user behavior may not accurately reflect user interests. Here, we consider two key scenarios. First, due to limited system recommendations, items matching user's certain interests may not have been interacted with, causing a lack of feedback, known as exposure bias. For example, a user with an interest in travel may like tourism posts on social media, but they may also be interested in photography tips that haven't appeared in their feed yet. Second, due to individual differences, users adopt different interaction strategies, leading to selection bias [10]. For instance, some users may primarily rate the shoes they purchase while leaving everyday items unrated, which does not imply a lack of interest in those items. As such, data bias makes it difficult for conventional methods to fully capture users' *underexplored preferences* without external knowledge, which may be overshadowed by dominant preferences in the observed data or may not yet appear, leading to narrow suggestions that cater only to dominant interests and placing users in "rabbit hole" [17, 19]. To mitigate this issue, we aim to investigate the task of diversified recommendation.

In this paper, we explore the feasibility of enhancing recommendation diversity while minimizing relevance loss by leveraging the domain knowledge provided by large language models (LLMs). As LLMs demonstrate unprecedented capability in zero-shot inference using world knowledge, our key insight is to leverage their expertise to analyze the rationale behind user historical interactions and uncover underexplored preferences, thus addressing the negative impact on diversity from data bias. For instance, a user who frequently browses travel and transportation posts on social media can be profiled as a travel enthusiast by the LLM, which can further infer potential interests in local cuisines or photography. In doing so, we strive to unbiasedly recover user preferences, thereby facilitating *diverse yet relevant* enhancements. Nevertheless, achieving this goal entails several challenges.

The first challenge is how to effectively leverage LLMs to capture a user's underexplored preferences from biased observations. Existing works using LLMs for diversified recommendations often fall short in systematic, fine-grained analysis of user preferences [20–22]. Some approaches directly match user history with items [20] or infer preferences at the category level [21]. We argue that coarse-grained matching is unsuitable for exploring latent preferences in diversity enhancement, as broad preferences may introduce noise, thereby diminishing relevance. To solve this, we design Tree of Preferences (ToP), which models user preferences from coarse to fine, to help the LLM better analyze the rationale behind user historical behaviors and improve the inference of latent interests.

The second challenge is how to leverage the uncovered preferences to guide diverse and relevant recommendations for users. A straightforward solution is to leverage LLMs to generate items in the embedding space [22], such as a two-step grounding paradigm [21, 23], but it suffers from suboptimal inference latency. Another is to have LLM rank items within the candidate set, either the full item set or a subset narrowed by external aids [20, 24], but its performance depends heavily on the size or quality of the candidates. To address this challenge, we adopt a data-centric approach, where candidate items matching latent user preferences are identified via the LLM. We then generate

synthetic interactions that best reflect user underexplored interests and integrate them into a general recommender for training. Moreover, we speed up the efficiency by dynamically selecting influential users during the optimization process. Our contributions are as follows:

- We propose ToP-Rec, a novel approach that explores diversified recommendation from a data-bias perspective, aiming to enhance diversity while maintaining relevance with expertise from LLMs.

- We design Tree of Preferences to model fine-grained user interests, serving as a vehicle for LLMs to uncover underexplored preferences from observed behaviors. Synthetic interactions are generated to supplement existing data, training a general recommender for diversified suggestions.

- Extensive experiments on three real-world datasets show that ToP-Rec achieves advantages in both diversity and relevance in most cases, with a dominant trade-off and efficient inference latency compared to baselines.

## 2 Preliminary

The recommendation diversity referred to in this paper measures the dissimilarity of items recommended to an individual user [16]. A closely related but orthogonal concept is the novelty of recommendations [25], also referred to as serendipity, popularity bias, or even diversity in some works [15, 26–28]. For consistency, we refer to this concept as novelty in this paper. Novelty measures the proportion of long-tail or unpopular items among the recommendations for different users [15, 25]. Given the fundamental difference between diversity and novelty, this paper focuses on enhancing recommendation diversity, excluding novelty from its scope.

**Diversity-relevance trade-off in recommendation.**    The trade-off between recommendation diversity and relevance has been extensively studied in prior work [1, 6, 8, 16]. In essence, this phenomenon arises from the fact that introducing dissimilarity may lead to additional noisy recommendations that are irrelevant to the user. Nevertheless, we emphasize that dissimilar items are not necessarily irrelevant; they may harbor implicit correlations by sharing common aspects aligned with user potential preferences. For example, a list comprising sneakers, sports socks, a smartwatch, and a yoga mat is generally more relevant to fitness enthusiasts than a list comprising office supplies, home decor, and kitchenware. Therefore, if latent user preferences can be captured, enhancing diversity while preserving relevance becomes feasible. In this work, we investigate the balance between diversity and relevance in recommender systems with the aid of LLMs. With this in mind, we next present a formal definition of the target problem.

**Problem formulation.**    Let $\mathcal{U}$ denote the set of users and $\mathcal{I}$ denote the set of items. $\mathcal{R} \subseteq \mathcal{U} \times \mathcal{I}$ is the interaction set between users and items. In this paper, we assume each user $u \in \mathcal{U}$ is associated with an attribute set $\mathcal{A}_u$, and each item $i \in \mathcal{I}$ is associated with an attribute set $\mathcal{A}_i$, where attributes are described by natural languages. In this context, a recommender system aims to maximize the diversity among the items suggested to each user while promoting their relevance. Formally, we define this as the diversity-relevance aware recommendation problem.

**Problem 1** (Diversity-relevance aware recommendation). *Given a user set $\mathcal{U}$ with attributes $\{\mathcal{A}_u | u \in \mathcal{U}\}$, an item set $\mathcal{I}$ with attributes $\{\mathcal{A}_i | i \in \mathcal{I}\}$, and an interaction set $\mathcal{R} \subseteq \mathcal{U} \times \mathcal{I}$, the diversity-relevance aware recommendation aims to learn a scoring function $f : \mathcal{U} \times \mathcal{I} \to \mathbb{R}$ with respect to the following two objectives:*

***Objective 1: relevance.*** *The recommender aims to maximize the relevance of the recommended items for each user $u$, i.e., $\max_{X \subseteq \mathcal{I}} \mathsf{Rel}(u, \mathcal{X})$, where $\mathcal{X} \subseteq \mathcal{I}$ is the set of recommended items for user $u$, typically comprising items with the highest scores assigned by the recommender. $\mathsf{Rel}(u, \mathcal{X})$ is a relevance metric for item set $\mathcal{X}$ and user $u$, such as recall.*

***Objective 2: diversity.*** *The recommender aims to enhance the diversity of the recommended items for each user $u$, i.e., $\max_{X \subseteq \mathcal{I}} \mathsf{Div}(\mathcal{X})$, where $\mathsf{Div}(\mathcal{X})$ is a diversity metric for item set $\mathcal{X}$, such as category entropy.*

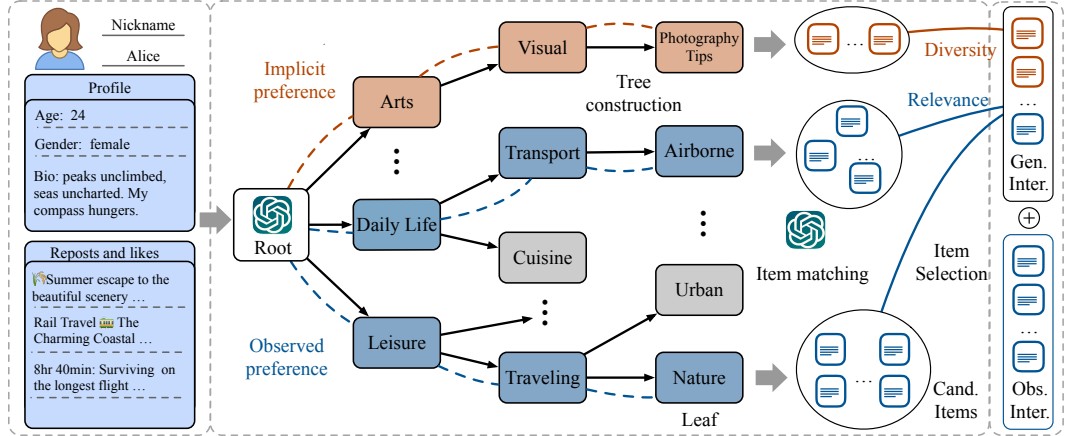

Figure 1: Illustration of our approach: **T**ree **of P**references for diversified **Rec**ommendation (ToP-Rec). Given "Alice" with her attributes and interacted items, ToP-Rec infers her rationale along the constructed ToP by prompting LLMs to comprehensively uncover her preferences until reaching the leaf nodes. Items aligned with her preferences are then matched, and synthesized user-item interactions concerning diversity and relevance are generated and integrated with the observed interactions. The combined data enables the recommender to offer diversified suggestions.

# 3 Methodology

In this section, we detail the design of ToP-Rec. We begin with an overview, then introduce the two steps of ToP-Rec, where a user's latent preferences are first unveiled (§ 3.2) to guide the generation of synthetic interactions (§ 3.3). We further discuss a strategy for scaling up our approach in § 3.4.

## 3.1 Overall framework

Given users' textual attributes and their interaction history, ToP-Rec aims to generate diverse and relevant recommendation lists for them. We provide an illustration in Figure 1:

(1) ToP-Rec first uncovers latent user preferences by constructing ToP (black arrows) and inferring latent preferences through systematic reasoning on ToP (dashed lines);

(2) ToP-Rec generates synthetic interactions that recover the true preference distribution by identifying items aligned with inferred preferences and selecting those that enhance diversity and relevance.

(3) Synthetic interactions are integrated with the original, and a general recommender is trained on the combined data. To scale up, ToP-Rec dynamically selects users for synthetic interaction generation.

## 3.2 Uncovering underexplored preferences

Despite LLMs' domain knowledge, uncovering a user's unexplored preferences from biased historical interactions remains challenging. Existing works use LLMs to analyze user preferences based on history, yielding promising results [20, 21]. However, these approaches lack a systematic analysis of fine-grained user interests, which are essential for providing diverse recommendations while maintaining relevance. Inspired by ToT [29, 30], we propose establishing a hierarchical user preference structure to enable the LLM to reason the rationale behind user historical behaviors from coarse to fine, ultimately uncovering a user's unexplored preferences.

**Constructing Tree of Preferences over items.** First, we discuss how to construct a hierarchy tree representing user preferences. Essentially, this process involves progressively dividing the *space* of user preferences over items. Figure 1 shows an example partition in a social media context, where user interest in travel posts is initially separated from leisure, and subsequent partitions lead to fine-grained preferences like nature or urban tours. Such a hierarchical structure serves as a vehicle, embodying the LLM's systematic reasoning of the user's rationale behind their behavior.

Starting from the root node, we instruct the LLM to partition the preference space progressively until a fine-grained division is achieved. To ensure sufficient knowledge of item distribution, we use a $k$-means-based method to sample a smaller, text-rich subset of items $\mathcal{S} \subset \mathcal{I}$ for the LLM. Specifically, we first encode each item's text-based attributes using a pretrained language model (*e.g.*, BERT [31]). To avoid excessive semantic redundancy, we then perform $k$-means clustering on the item embeddings and randomly sample items from each cluster. The sampled items are aggregated as the item subset $\mathcal{S}$. Then, the construction of ToP can be formally represented as

$$\mathcal{T}(\mathcal{V}, \mathcal{E}) = \mathsf{LLM}(\mathsf{Prompt}_{\mathsf{ToP}}(\mathcal{S})), \tag{1}$$

where $\mathcal{V}$ denotes the node set of hierarchical preferences, $\mathcal{T}$ denotes the tree of preferences, and $\mathcal{E} \subset \mathcal{V} \times \mathcal{V}$ denotes the edge set. Each node $v \in \mathcal{V}$ represents a kind of user preference generated by LLM, denoted by a token sequence $[\mathsf{Token}_0, \mathsf{Token}_1, \dots]$. Each edge $e \in \mathcal{E}$ represents a finer preference division from its parent node. $\mathsf{Prompt}_{\mathsf{ToP}}$ means the instructions for constructing ToP (see Appendix A.2). In practice, if the item set has a predefined categorization, it will also be provided as a reference in the prompt of ToP.

Note that the tree of preferences is constructed as a global, one-time structure shared across all users. This design stems from the insight that while individual preferences diverge, all users navigate the same underlying preference space shaped by item attributes.

**Capturing user latent preferences via rationale reasoning.** Next, we explain the systematic reasoning behind the user's behavior. Given their historical interactions, the LLM performs a top-down exploration of the hierarchical preferences, identifying the coarse-to-fine path that best matches the user's behavior (see dashed lines in Figure 1. This enables a systematic analysis of user rationale and fine-grained preferences. The LLM also reevaluates the exploration with respect to the user's rationale, checking for unobserved preferences not reflected in the interaction history.

Based on user interactions, a breadth-first search algorithm is used. Starting from the root node, the LLM selects preferences $v^l$ that best explain the user behavior at each level, storing the corresponding nodes. As it moves to the next $l+1$-level, only the stored nodes' children are activated, continuing until the leaf node. The LLM then summarizes the user's behavior rationale and revisits the exploration to check for unobserved preferences, which are added if found, following the same root-to-leaf path. Finally, the leaf nodes of all selected paths are returned. The process can be summarized as:

$$\{v_1, \dots v_n\} = \mathsf{LLM}(\mathsf{Prompt}_{\mathsf{PR}}(\mathcal{A}_u, \mathcal{R}_u)), \{v_1, \dots v_n\} \subset \mathcal{V}_{\mathsf{Leaf}}, \tag{2}$$

where $\{v_1, \dots, v_n\}$ denote the selected leaf nodes representing user complete preferences, $\mathcal{R}_u \subset \mathcal{R}$ denotes user $u$'s interactions, and $\mathsf{Prompt}_{\mathsf{PR}}$ is the instruction for preference reasoning.

### 3.3 Generating synthetic interactions

As we uncover underexplored preferences, the next challenge lies in generating unbiased interactions based on them. We adopt a data-centric approach, leveraging data augmentation to mitigate potential biases in the observed feedback while ensuring efficient inference latency. First, candidate items that align with user preferences are identified. Then, items that cover underexplored preferences while considering relevance are selected and incorporated into existing user interactions. The synthetic unbiased interactions are ultimately used to train a diversified recommender.

**Matching candidate items with Tree of Preferences.** To find candidate items matching the preferences of any user, we aim to assign items from the entire set to the corresponding leaf nodes in ToP (*cf.* Eq. (2)). To reduce repeated computation, we pre-assign each item to its best matching leaf node by providing the item's textual attributes $\mathcal{A}_i$ to the LLM, which identifies the suitable leaf node in ToP. Finding candidate items that match specific preferences can then be easily achieved. The pre-assignment of each item is represented as follows:

$$v_i = \mathsf{LLM}(\mathsf{Prompt}_{\mathsf{IM}}(\mathcal{A}_i)), v_i \in \mathcal{V}_{\mathsf{Leaf}}, \tag{3}$$

where $v_i$ denotes the assigned leaf node of item $i$ via LLM and $\mathsf{Prompt}_{\mathsf{IM}}$ denote the prompt for item matching. Note that this process can be completed once ToP is constructed (see § 3.2), and we present it here for clarity. To improve the load imbalance of leaves, we also incorporate refinement mechanisms after assignment. Please refer to Appendix A.2 for a detailed description.

**Data generation for debiasing user interactions.** Next, we select items that cover underexplored preferences, which are either overlooked or not yet manifested in the user's behavior. To do this, we calculate each item's contribution to diversity by measuring its impact on debiasing. Intuitively, selecting an item that reflects a latent preference with low (no) occurrence in the user's history has a higher impact. Thus, the diversity score of an item $i$ is defined as $s_{\mathsf{div}}(u, i) \propto 1/\mathsf{freq}_i$, where $\mathsf{freq}_i$ represents the frequency of the preference associated with item $i$ in $u$'s history. We also measure the relevance of each item by calculating its semantic alignment with the user: $s_{\mathsf{rel}}(u, i) = \langle \mathsf{Enc}(\mathcal{A}_u), \mathsf{Enc}(\mathcal{A}_i) \rangle$, where $\mathsf{Enc}(\cdot)$ denotes a pretrained language model (*e.g.*, BERT [31]) and $\langle \cdot, \cdot \rangle$ denotes cosine similarity. Finally, the overall score is computed as:

$$s(u, i) = (1 - \lambda) \cdot s_{\mathsf{rel}}(u, i) + \lambda \cdot s_{\mathsf{div}}(u, i), \tag{4}$$

where $\lambda$ is a hyperparameter used to balance relevance and diversity. For a given user, we calculate the scores of all candidate items and select those above a predefined threshold. To reduce bias and better reflect user interests, the selected items are added to the user's interaction history, obtaining synthetic interactions $\mathcal{R}'_u = \mathcal{R}^+ \cup \mathcal{R}_u$, where $\mathcal{R}^+$ represents the selected items. Finally, the synthetic interactions are used for training a general recommender, leading to diverse yet relevant performance.

### 3.4 Cost-efficient interaction generation

Limited by token throughput and LLM latency, generating interactions for every user can be costly. To address this, we propose a cost-efficient strategy that identifies influential users for interaction generation, balancing improvements with costs. Given the complexity of data and recommender designs, it is infeasible to design a static heuristic to distinguish user importance, so we dynamically quantify each user's influence based on the recommender's feedback during training. To be specific, the recommender is initially trained on the original interactions. As user influence varies during training, we backtrack parameter updates and compute user influence at fixed intervals. The most influential users are selected to generate synthetic interactions, which are integrated into the training set. This continues until the model reaches peak performance. To quantify user influence, we design a custom criterion based on gradient alignment, measuring each user's contribution by the alignment between their local gradient and the model's parameter trajectory.

Assume the recommender is optimized using Bayesian Personalized Ranking (BPR) loss* [32], defined as: $\mathcal{L} = -\sum_{u \in \mathcal{U}} \sum_{(u,i) \in \mathcal{R}_u} \sum_{(u,j) \notin \mathcal{R}_u} \ln \sigma(\hat{y}_{ui} - \hat{y}_{uj})$, where $\hat{y}$ denotes similarity scores. The local loss incurred by user $u$ is: $\ell(u; \theta) = -\sum_{(u,i) \in \mathcal{R}_u} \sum_{(u,j) \notin \mathcal{R}_u} \ln \sigma(\hat{y}_{ui} - \hat{y}_{uj})$. Based on this, we define a user's influence in the gradient descent process over $k$ steps.

**Definition 1** ($k$-step influence). *Given the local gradient $\nabla \ell(u; \theta)$ of user $u$, and the gradient descent trajectory of model parameters $\{\theta^0, \dots \theta^t\}$ backward from step $t$, the $k$-step influence of user $u$ is defined as $\mathsf{Inf}_u = \sum_{i=t-k}^{t} \langle \nabla \ell(u; \theta^{i-1}), \theta^i - \theta^{i-1} \rangle$.*

With numerous users or high-dimensional gradients, the computational cost of user influence increases. We apply gradient dimension reduction [33] and group users to compute the group influence. More theoretical and empirical analyses are provided in Appendix A.2 and A.4.

**Discussion with existing work.** (1) Conventional diversified recommendations adopt various solutions to capture user preferences, such as uncertain masking [6], contrastive context learning [18], and user-category matching [9], which rely solely on observed data. However, due to inherent data bias, they may fail to fully capture preferences. In contrast, our approach moves beyond the scope of observed data, leveraging world knowledge from LLMs to reason about user rationale, offering greater potential to enhance diversity. (2) LLM-based diversified recommenders propose reranking solutions [20], or use LLM fine-tuning [22] to capture user preferences for item genres [21]. However, these approaches focus on a coarse category level, which can lead to noisy recommendations and affect accuracy. In contrast, our ToP models user preferences in a coarse-to-fine manner, facilitating nuanced reasoning over user rationale for better diversity and relevance.

## 4 Experiments

In this section, we evaluate the performance of ToP-Rec through extensive experiments. Due to space limitations, please refer to Appendix A.3 and A.4 for more experimental settings and results.

---

*Note that our method can be applied to other types of losses, such as binary cross-entropy loss.

### 4.1 Experimental setup

**Datasets.** We use the Twitter [11], Weibo [12], and Amazon [14] datasets. Twitter and Weibo are social network datasets with user attributes (e.g., username, location, bio) and posts as items, including attributes like retweet counts and content. User feedback consists of likes and retweets. Amazon is an e-commerce dataset, where we combine seven categories as in [34]. For each dataset, we follow [16] to extract a subset of the original, then drop items with missing attributes and apply 10-core filters (5-core for Twitter), resulting in a dataset that retains relatively informative and active entities. For each user, we split their interactions into train, validation, and test sets with a ratio of 0.6:0.2:0.2. See Appendix A.3 for details and statistics of datasets.

Following prior work [24], we also enhance the quality of user and item textual attributes in these datasets using LLMs. Specifically, we leverage an LLM to generate user summaries based on their profile and the textual attributes of previously interacted items, as well as to generate item summaries from their textual attributes. By generating these summaries, we aim to reduce noise, redundancy, and inconsistencies in the original text, leading to improved summaries that retain key semantic information.

**Evaluation metrics.** To evaluate the relevance of recommendations, we follow [16] and adopt the metric Recall@$k$ (R@$k$), indicating the proportion of relevant items retrieved in the top-$k$ recommendation list. To assess diversity, we use the Category-Entropy@$k$ (CE@$k$), which measures the distribution of different categories within the top-$k$ list. We report $k = 50$ and $100$ in this work.

**Baselines.** We adopt nine baselines to compare with the proposed approach, categorized into three types: (1) Heuristic methods: Random, MMR [35], and DPP [36]; (2) Conventional diversity-enhancing methods: Box/LCD-UC [6] and CDM [18]; (3) LLM-based diversified recommender: LLM4Rerank-A/LLM4Rerank-AD [20] and LLMRec-MMR [24]. Detailed descriptions and configurations for all baselines are provided in Appendix A.3.

**Implementation details.** We implement LightGCN with 2 hidden layers and a hidden size of 32, which is optimized using Adam optimizer with a learning rate of 5e-3. We also evaluate the performance of ToP-Rec on other backbones (see Appendix A.4). We employ a random negative sampling with a 1:50 ratio and use early stopping. For hyperparameters affecting diversity and relevance, we search the number of selected leaves in $[4, 7]$ (step size 1), number of augmentations per user in $[3, 9]$(step size 2), and the item sampling weight $\lambda$ in $[0.2, 0.8]$ (step size 0.2). We utilize Qwen2.5-32B-Instruct [37] to complete tasks involving LLMs. To ensure fairness, we employ the same LLM for our approach and all baselines involving LLMs. Experiments are repeated 5 times to report the average performance with standard deviation.

All experiments are conducted on a machine of Ubuntu 20.04 system with AMD EPYC 7763 (756GB memory) and NVIDIA RTX3090 GPU (24GB memory). All models are implemented in PyTorch version 2.5.1 with CUDA version 11.8 and Python 3.10.15. Our code is publicly available at `https://github.com/xxx08796/ToP_Rec_NIPS`.

### 4.2 Evaluation of performance

We first evaluate ToP-Rec's overall performance in terms of diversity and relevance. For fairness, we select a balanced result for methods with adjustable hyperparameters and run other baselines with their original settings. For reranking methods like MMR and DPP, we use 10 times the top-$k$ value as the candidate list, and for LLM4Rerank-A and LLM4Rerank-AD, we use twice the top-$k$ value due to instability. Table 1 presents average recall and category-entropy comparisons, revealing several insights: (1) ToP-Rec dominates in most cases, with only one suboptimal result, showing its advantage in both diversity and relevance. (2) LLM-based methods perform relatively well in relevance, but improvement in diversity is limited, likely due to a lack of fine-grained preference analysis, leading to redundant item selections. (3) LCD-UC and Box struggle with high relevance, as box embeddings increase similarity with irrelevant items. (4) Heuristics like MMR are hard to achieve a balance, excelling in one aspect while underperforming in another, as observed in [16].

Table 1: Comparison of performance on diversity (R@$k$) and relevance (CE@$k$). $^*$ denotes the backbone model, and $^+/^-$ indicates performance improvements or declines compared to the backbone. The optimal performance is in bold, and the second-best performance is underlined.

| | Twitter | | | | Weibo | | | | Amazon | | | |
|---|---|---|---|---|---|---|---|---|---|---|---|---|
| | R@50 | R@100 | CE@50 | CE@100 | R@50 | R@100 | CE@50 | CE@100 | R@50 | R@100 | CE@50 | CE@100 |
| LightGCN$^*$ | 0.0567 | 0.0830 | 1.2841 | 1.3413 | 0.1052 | 0.1669 | 0.9905 | 1.0763 | 0.1362 | 0.2105 | 0.5004 | 0.5609 |
| Random | 0.0494$^-$ | 0.0730$^-$ | 1.2954$^+$ | 1.3475$^-$ | 0.0988$^-$ | 0.1577$^-$ | 0.9995$^+$ | 1.0801$^+$ | 0.1356$^-$ | 0.2062$^-$ | 0.5184$^+$ | 0.5792$^+$ |
| MMR | 0.0540$^-$ | 0.0790$^-$ | 1.3078$^+$ | 1.3550$^-$ | 0.0990$^-$ | 0.1578$^-$ | 1.0081$^+$ | 1.1005$^+$ | 0.1371$^+$ | 0.2115$^+$ | 0.5141$^-$ | 0.5755$^+$ |
| DPP | 0.0467$^-$ | 0.0765$^-$ | 1.3048$^+$ | 1.3532$^-$ | 0.0963$^-$ | 0.1530$^-$ | **1.0362$^+$** | 1.1150$^+$ | 0.1283$^-$ | 0.2035$^-$ | 0.5181$^+$ | 0.5748$^+$ |
| CDM | 0.0562$^-$ | 0.0814$^+$ | 1.2986$^+$ | 1.3461$^-$ | 0.1014$^-$ | 0.1620$^-$ | 1.0018$^+$ | 1.0912$^+$ | 0.1349$^-$ | 0.2103$^-$ | 0.5228$^+$ | 0.5816$^+$ |
| Box | 0.0527$^-$ | 0.0741$^-$ | 1.2844$^+$ | 1.3407$^-$ | 0.0996$^-$ | 0.1587$^-$ | 1.0238$^+$ | 1.1034$^+$ | 0.1228$^-$ | 0.2019$^-$ | 0.5186$^+$ | 0.5844$^+$ |
| LCD-UC | 0.0517$^-$ | 0.0768$^-$ | 1.3154$^+$ | 1.3784$^-$ | 0.1038$^-$ | 0.1625$^-$ | 1.0211$^+$ | 1.0956$^+$ | 0.1295$^-$ | 0.2065$^-$ | 0.5202$^+$ | 0.5842$^+$ |
| LLMRec-MMR | 0.0558$^-$ | 0.0820$^-$ | 1.3056$^+$ | 1.3551$^-$ | 0.1041$^-$ | 0.1662$^-$ | 1.0246$^+$ | 1.1182$^+$ | 0.1363$^+$ | 0.2113$^+$ | 0.5177$^+$ | 0.5836$^+$ |
| LLM4Re-A | 0.0562$^-$ | 0.0827$^-$ | 1.2855$^+$ | 1.3424$^-$ | 0.1032$^-$ | 0.1656$^-$ | 0.9891$^-$ | 1.0745$^-$ | 0.1359$^-$ | 0.2049$^-$ | 0.5028$^+$ | 0.5863$^+$ |
| LLM4Re-AD | 0.0560$^-$ | 0.0822$^-$ | 1.2864$^+$ | 1.3466$^-$ | 0.1044$^-$ | 0.1652$^-$ | 1.0001$^+$ | 1.0823$^+$ | 0.1332$^-$ | 0.2042$^-$ | 0.5131$^+$ | 0.5827$^+$ |
| ToP-Rec | **0.0586$^+$** | **0.0841$^+$** | **1.3275$^+$** | **1.3852$^+$** | **0.1054$^+$** | **0.1667$^-$** | 1.0333$^+$ | **1.1369$^+$** | **0.1380$^+$** | **0.2120$^+$** | **0.5298$^+$** | **0.5902$^+$** |

**Relevance-diversity trade-off.** To further demonstrate the robustness of our proposed method, we evaluate the relevance-diversity trade-off of ours and the baselines that support adjustments to balance this trade-off, including Random, MMR, DPP, CDM, LCD-UC, and LLMRec-MMR. We tune the hyperparameters of each method to explore their trade-offs, selecting the best three trade-off points for each method and visualizing those for the Twitter dataset in Figure 2. The upper-right corner represents the ideal performance, with higher recommendation relevance and diversity. The result shows that our approach achieves the best trade-off compared with other baselines. In particular, while the diversity is enhanced with a larger margin, the relevance under our approach is consistently improved upon the backbone LightGCN (shown by a star mark). We also compare the trade-offs on Weibo and Amazon, which can be found in the Appendix A.4.

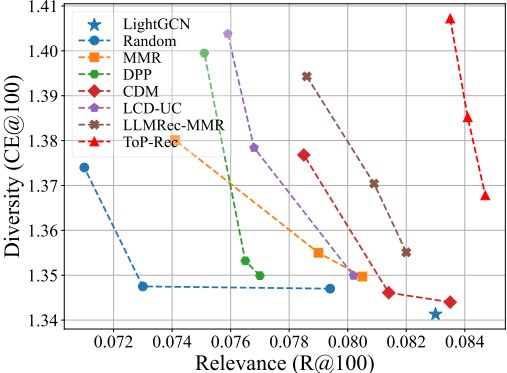

Figure 2: Diversity-relevance trade-off comparison. The upper-right represents the ideal.

**Inference latency.** To evaluate our method's efficiency, we compare its inference time with other methods, measuring the average time to generate a recommendation list for a user. Figure 3(a) shows that our method achieves similar latency to traditional methods like CDM and LCD-UC, while outperforming the reranking method MMR. Notably, it also shows a significant advantage over LLM-based method LLM4Rerank. In summary, our method can optimize diversity-relevance performance while maintaining efficient inference latency compared to baselines.

## 4.3 Ablation study

We conduct ablation studies to evaluate the effectiveness of each component in our approach, including four variants: (1) w/o Div and (2) w/o Rel: We ignore the item's contribution to diversity or relevance in item selection (*cf.* Eq.(4)); (3) w/o US: We discard the influential user selection (*cf.* § 3.4) and random select user for augmentation; (4) w/o PR: We avoid LLM to infer user preferences, instead randomly selecting leaf nodes (*cf.* Eq.(2)). Figure 3(b) visualizes their changes in diversity and relevance, ordered by relevance in descending order. First, w/o Div and w/o Rel outperform Ours in relevance and diversity, respectively, but perform poorly in the other aspect due to considering only one factor. Second, w/o US improves both aspects but remains weaker than Ours, showing that augmentation on influential users boosts performance. Finally, w/o PR performs worst in relevance, indicating that ignoring user interests increases the risk of irrelevance.

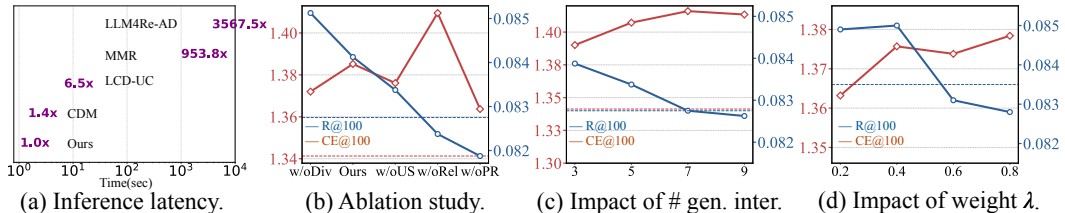

(a) Inference latency.  (b) Ablation study.  (c) Impact of # gen. inter.  (d) Impact of weight $\lambda$.

Figure 3: (a) Comparison of average time to generate recommendations; (b) Evaluation of each component in ToP-Rec; (c) and (d) Impact of generated interactions per user and selection weight $\lambda$. We use dashed lines to represent the performance of the backbone recommender.

## 4.4 Hyperparameter analysis

Next, we evaluate the influence of important hyperparameters in our approach, which impact the diversity-relevance performance. These mainly include: (1) the weight $\lambda$ for item selection. (2) the number of generated interactions per user. (3) the number of selected leaf nodes per user. We tune the weight $\lambda$ among $\{0.2, 0.4, 0.6, 0.8\}$ and number of generated interactions among $\{3, 5, 7, 9\}$. The results in Figure 3(d) show that as the weight $\lambda$ for item selection increases, diversity rises while relevance declines. This is because a higher weight increases the focus on diversity, raising the possibility of irrelevance. Figure 3(c) shows that as the number of generated interactions increases, relevance decreases, while diversity initially rises and then stabilizes. This is due to the decline in item relevance, which reduces recall, while the contribution to diversity also diminishes, stabilizing its growth. Due to page limitations, we analyze the impact of other hyperparameters in the appendix.

## 5 Literature Review

**Diversified recommendation.** Research on diversified recommendation has a well-established history. Early works concentrate on re-ranking diversification [35, 36, 38–42]. These approaches often leverage greedy solutions to balance utility and diversity [35, 39, 40], or employ Determinantal Point Processes [43] to generate diverse recommendations by measuring set diversity [36, 41, 42]. Recently, more complex diversified re-ranking methods have been proposed, such as introducing a user-behavior model to maximize knowledge diversity [17], or employing graph clustering to capture user interests and sub-models to assess diversity [1]. Unlike these post-matching works, our framework directly incorporates diversification in the matching stage. Many recent works also focus on the matching stage, with some built upon Bayesian graph convolutional neural networks [8], multi-vector representations [7], two-stage category-item matching [9], or rebalanced sampling techniques [16]. Compared to them, we propose a universal pipeline for use with generic recommender backbones, instead of designing a specific model. Our work is most similar to [6, 18], where they propose a general framework for diversified recommendation via box embedding and uncertainty masking[6], or knowledge distillation learning from MMR [18]. A common limitation of traditional methods is their reliance on observed data, which hinders overcoming diversity decline due to inherent data bias. In contrast, our approach moves beyond the scope of observed data, leveraging world knowledge from LLMs to reason about the user rationale, offering greater potential to enhance diversity.

**LLM-based recommendation.** With the impressive capabilities demonstrated by LLMs [44–47], a growing body of work explores their application in recommender systems [48–50]. Early research primarily focuses on leveraging LLMs to enhance the relevance of recommendations [24, 51–54], utilizing the reasoning abilities of LLMs to analyze potential user interests and generate tailored recommendations. Recently, LLM-based recommendations have expanded beyond relevance, with a growing focus on other performance aspects [55], especially on recommendation diversity [20–22]. To mitigate the homogeneity issue in LLM-based recommendations, the decoding strategy in [22] integrates a text-free assistant model to refine the token scores. However, its effectiveness depends on the assistant model's quality; if the model provides poor suggestions, it may lead to irrelevant or low-quality recommendations. DLCRec [21] introduces a framework for diversity control in LLM-based recommendations by breaking down the recommendation task into three sub-tasks; LLM4Rerank [20] proposes a LLM-based reranking approach that leverages a graph structure to represent accuracy, diversity, and fairness in reranking, allowing for the refinement of the final recommendations. Despite some improvement in diversity, these approaches lack fine-grained user preference analysis and item

space partitioning, leading to a higher risk of irrelevant recommendations and affecting accuracy. In contrast, Tree of Preferences hierarchically refines user preferences from coarse to fine, enabling the LLM to uncover underexplored user preferences through nuanced reasoning, thereby facilitating diverse and relevant recommendations.

**Influential data selection.**    Existing research on influential data selection aims to estimate the influence of individual or sets of training records on model performance and select the most influential ones [33, 56–58]. In this paper, we primarily discuss two widely used approaches: influence function-based methods [56, 59–61] and gradient descent tracing-based methods [33, 62–64]. While influence functions provide insights into how perturbations to certain parts of the training data affect model behavior [65], the computation of the inverse-Hessian limits the effectiveness of the selection process [66]. Furthermore, existing work has pointed out that traditional influence functions may fail on certain types of data and model architectures. For instance, when applied to Graph Neural Networks (GNNs) [67–70], computing influence requires analyzing the mutual interactions between nodes [71–75], which adds significant computational overhead when using influence functions for user selection in many GNN-based recommendation models [76]. On the other hand, studies [62] utilize first-order approximation to estimate the influence of a training sample on model performance under stochastic gradient descent. [33] extends this assumption to the Adam optimizer. The methods in [63, 64] are closest to ours, calculating the alignment between the local gradient of pretrain samples and the downstream loss gradient. However, the downstream loss gradient may not align with the actual parameter update during fine-tuning. In contrast, we calculate the alignment between the local gradient of a sample and the actual parameter update trajectory, ensuring a more accurate measure.

# 6    Broader impacts

Our approach improves recommendation diversity by uncovering underexplored user preferences through LLMs, addressing two important societal challenges in recommendation systems. First, it reduces information redundancy by diversifying recommendations, helping users break free from repetitive suggestions and explore niche topics, which reduces the impact of algorithmic echo chambers. Second, by diversifying recommendations, users are given more opportunities to encounter content they may not have come across but are likely to be interested in, thereby fostering the diversity of societal culture. These improvements contribute to more balanced, inclusive recommendation systems that prioritize user-driven discovery over algorithmic determinism.

# 7    Conclusion

In this paper, we explore diversified recommendation from a data-bias perspective, identifying two key scenarios that may introduce bias affecting diversity. To address this, we propose ToP-Rec, which leverages external knowledge from LLMs to complement data bias. We construct Tree of Preferences to model user preferences from coarse to fine, helping the LLM analyze user behaviors and improve latent interest inference. To ensure efficient recommendations, candidate items matching latent preferences are identified via the LLM, and synthetic data is generated through a relevance-diversity-aware strategy for training. Additionally, we introduce a dynamic user selection mechanism to reduce costs by selecting influential users based on gradient feedback. We extensively evaluate the performance of ToP-Rec on three real-world datasets, comparing it to nine competitive baselines. The results demonstrate that ToP-Rec outperforms in most cases, achieving second-best performance in others, with the optimal trade-off between diversity and relevance and efficient inference latency.

## Acknowledgments and Disclosure of Funding

This work is supported in part by Zhejiang Provincial Natural Science Foundation of China (Grant No. LMS25F020012), the Hangzhou Joint Fund of the Zhejiang Provincial Natural Science Foundation of China under Grant No.LHZSD24F020001, Zhejiang Province High-Level Talents Special Support Program "Leading Talent of Technological Innovation of Ten-Thousands Talents Program" (No.2022R52046), the Fundamental Research Funds for the Central Universities (2021FZZX001-23), the advanced computing resources provided by the Supercomputing Center of Hangzhou City University, and CIPSC-SMP-Zhipu Large Model Cross-Disciplinary Fund.

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

# A Appendix

## A.1 Notations

Table 2: Description of major notations, ordered by appearance.

| Notation | Description |
|---|---|
| $\mathcal{U}, \mathcal{I}, \mathcal{R}$ | user set, item set, interaction set |
| $u, i, \mathcal{A}$ | user, item, attribute set of user or item |
| $f$ | scoring function |
| $\mathcal{X}, \mathsf{Rel}(\cdot, \cdot), \mathsf{Div}(\cdot)$ | set of recommended items, relevance metric, diversity metric |
| $\mathcal{S}$ | set of sampled items for constructing ToP |
| $\mathsf{Prompt}, \mathsf{LLM}(\cdot)$ | input prompt for LLM, output of LLM |
| $\mathcal{T}, \mathcal{V}, \mathcal{E}$ | tree, node set of $\mathcal{T}$, edge set of $\mathcal{T}$ |
| $v, e$ | node of $\mathcal{T}$, edge of $\mathcal{T}$ |
| $\mathcal{R}_u$ | user $u$'s interactions |
| $\langle \cdot, \cdot \rangle, \mathsf{Enc}(\cdot)$ | cosine similarity, pretrained language encoder |
| $\mathsf{freq}_i$ | frequency of the preference associated with item $i$ in user $u$'s history |
| $s(u, i), \lambda$ | item selection score, hyperparameter |
| $\mathcal{R}', \mathcal{R}^+$ | synthetic interactions, generated interactions from selected items |
| $\mathcal{L}, \theta$ | loss of recommender, parameter of recommender |
| $\ell(\cdot; \cdot), \nabla$ | user local loss, gradient |
| $\mathsf{Inf}_u$ | influence of user $u$ |

## A.2 Supplementary to the method

**Illustration of prompts.** We provide an illustration of the prompts used to complete the essential processes of ToP-Rec, as summarized in Figure 4.

**Details of ToP refinement.** To find candidate items matching the preferences of any user, we preassign every item to a leaf node in ToP (*cf.* Eq.(3)). However, this assignment may lead to slight load imbalance, with varying numbers of items assigned to different leaf nodes. Empirically, we observe that the imbalance can carry over into the recommended items. Therefore, after assigning all items, we introduce a refinement mechanism to improve load balance, involving two operations.

(1) Merge. Underloaded sibling leaves are merged to create a load-balanced leaf. This is achieved by integrating the assigned items and instructing the LLM to summarize their corresponding preferences, thus forming a merged leaf.

(2) Split. Leaves with excessive load are split into several load-balanced leaves. Given an excessively loaded leaf, the LLM is prompted to further refine the corresponding preference into several finer-grained ones, reassigning the items accordingly.

In practice, underloaded and excessively loaded leaves are identified using two predefined lower and upper thresholds, enabling self-stabilization through refinement. To prevent infinite refinement, we set the maximum number of operations as 10.

**Theoretical analysis of user selection.** To quantify user influence based on gradient alignment, we assess the alignment between each user's local gradient and the model's parameter trajectory during gradient descent (see Definition 1). The intuition is that the stronger the alignment between a user's optimization direction and that of the recommender model, the greater their contribution to the model's improvement. Here, we present the theoretical analysis supporting our approach, starting with a theorem that establishes the upper bound of the convergence step in gradient descent (GD).

**Theorem A.1.** *Let $\mathcal{L}(\theta)$ be a convex loss function with learnable parameter $\theta$, and let $\theta^* \in \arg\min_\theta \mathcal{L}(\theta)$ denote the optimal parameter. Define the initial error $e_0 := \mathcal{L}(\theta^0) - \mathcal{L}(\theta^*)$, and let $\nabla \mathcal{L}(\theta^t)$ denote the gradient at optimization step $t$. Assume $\mathcal{L}$ is $\beta$-Lipschitz smooth. Then the number*

> **Example prompts used for ToP construction, preference reasoning, and item matching.**
>
> **Notation**: Prompt$_{\text{ToP}}$.
> **Usage**: ToP construction.
> **Content**: You are an expert system designed to classify and organize user preferences based on the following information of item examples. Your task is to generate a multi-level tree structure for user preferences, called Tree of Preferences (ToP), progressively refining them from broad to specific preferences. Each node represents a type of user preference, and each edge signifies a finer preference division from its parent node.
> Constraints:
> - Each node should be divided into 3-5 finer preferences (branches), except for leaf nodes.
> - Use diverse preference-dividing criteria at each level.
> - Nodes must represent clear, actionable preferences.
> · · ·
> Items samples: {sampled_items_information}.
> Return ToP in the following format: {ToP_format}.
>
> **Notation**: Prompt$_{\text{PR}}$.
> **Usage**: Preference reasoning.
> **Content**: You are an expert system designed to capture user latent preferences through rationale reasoning. You will be provided with the user's profile, observed interactions with items, and a multi-level tree structure representing different user preferences, organized from broad to specific preferences (Tree of Preferences, ToP). Using the profile and interactions, your task is to identify the user's latent preferences by exploring the ToP top-down, following the coarse-to-fine paths that best match the user's behavior and reasoning the rationale behind their actions, while identifying any unobserved preferences not reflected in the history.
> Constraints:
> - Perform a breadth-first search. At each level, select the preferences that best match the user behavior, storing the corresponding nodes.
> - For subsequent levels, activate only the child nodes of the stored nodes at the previous level, continuing the selection until reaching final level.
> - Reevaluate the exploration to identify any unobserved preferences, adding them if found, following the root-to-leaf path.
> - Control the total number of selected paths as {number_of_paths}. The final output should be the leaf nodes of all selected paths.
> · · ·
> User profile: {user_profile}.
> User historical interactions: {user_interactions}.
> ToP: {ToP_content}.
> Return selected leaf nodes in the following format: {leaf_nodes_format}, along with a concise explanation of each selection in the format of {reason_format}.
>
> **Notation**: Prompt$_{\text{IM}}$.
> **Usage**: Item matching.
> **Content**: You are an expert system designed to match items with user preferences. You will be provided with items and their specific information, along with a multi-level Tree of Preferences (ToP) representing user preferences, structured from broad to specific divisions, where each node represents a preference type and each edge further refines the preferences. Your task is to find the most relevant preference for each item in ToP, moving through the levels of ToP based on the item's information, and return the final leaf node.
> Constraints:
> - At each level, select only the most appropriate node from the available options.
> - For the next level, select only from the child nodes of the node selected in the previous level.
> · · ·
> ToP: {ToP_content}.
> Items information: {items_information}.
> Return the selected leaf node in the following format: {leaf_node_format}.

Figure 4: Illustration of the prompts used in this work.

Table 3: Dataset statistics.

| Dataset | # Interactions | # Users | # Items | % Density |
|---------|----------------|---------|---------|-----------|
| Twitter | 40,223 | 2,118 | 7,199 | 0.2639% |
| Weibo | 661,783 | 14,663 | 5,711 | 0.7903% |
| Amazon | 134,857 | 3,598 | 8,747 | 0.4285% |

*of steps $T$ required to achieve $\mathcal{L}(\theta^T) - \mathcal{L}(\theta^*) \leq \varepsilon$ satisfies:*

$$T \leq \frac{2\beta(e_0 - \varepsilon)}{\min_{0 \leq t \leq \mathcal{B}} \|\nabla\mathcal{L}(\theta^t)\|^2}, \tag{5}$$

*where $\mathcal{B} = \mathcal{O}(\frac{1}{\varepsilon})$.*

*Proof.* Since $\mathcal{L}$ is a convex function with $\beta$-Lipschitz smoothness, we can obtain the following inequality: $\mathcal{L}(\theta') \leq \mathcal{L}(\theta) + \nabla\mathcal{L}(\theta)^T(\theta' - \theta) + \frac{\beta}{2}\|\theta' - \theta\|^2$. Suppose the learning rate is $\alpha$, we substitute the gradient descent update $\theta^{t+1} = \theta^t - \alpha\nabla\mathcal{L}(\theta^t)$:

$$\mathcal{L}(\theta^{t+1}) \leq \mathcal{L}(\theta^t) - \alpha\|\nabla\mathcal{L}(\theta^t)\|^2 + \frac{\alpha^2\beta}{2}\|\nabla\mathcal{L}(\theta^t)\|^2. \tag{6}$$

Choosing the optimal learning rate $\alpha = \frac{1}{\beta}$ (maximizing the descent under smoothness), we can obtain:

$$\mathcal{L}(\theta^{t+1}) \leq \mathcal{L}(\theta^t) - \frac{1}{2\beta}\|\nabla\mathcal{L}(\theta^t)\|^2. \tag{7}$$

The decrease in the objective function per iteration has the lower bound:

$$\mathcal{L}(\theta^t) - \mathcal{L}(\theta^{t+1}) \geq \frac{1}{2\beta}\|\nabla\mathcal{L}(\theta^t)\|^2. \tag{8}$$

Since each epoch $t$ reduces the error by at least $\frac{\|\nabla\mathcal{L}(\theta^t)\|^2}{2\beta}$, and from [77] we know that given $\varepsilon$, the convergence rate of GD method is $\mathcal{O}(\frac{1}{\varepsilon})$, thus the minimum number of epochs $T$ satisfies $T \leq \frac{2\beta(e_0-\varepsilon)}{\min_{t \leq \mathcal{B}} \|\nabla\mathcal{L}(\theta^t)\|^2}$, where $\mathcal{B} = \mathcal{O}(\frac{1}{\varepsilon})$. $\square$

Theorem 1 demonstrates the inverse relation between the convergence step and the gradient norm, which inspires our definition of the user influence. We use the inner product summation to measure the user influence, which essentially prioritizes users whose gradients exhibit persistent positive projection onto the global update trajectory. Users with high influence contribute more significantly to increasing the gradient norm's magnitude, thereby enhancing the training efficiency and convergence performance according to Theorem 1.

### A.3   More experiment settings

**Dataset details.**   The statistics of the datasets used in this work are summarized in Table 3.

- **Twitter** [11]: This dataset is originally collected from Twitter for bot detection. We remove bot users, considering human tweet user as user and tweet as item. We consider user behaviors such as likes and retweets as interactions. User attributes include name, description, location, and more, and item attributes include content, retweet count, etc.
- **Weibo** [12]: This dataset is collected from Weibo, one of China's largest social media platforms. We treat microblog posts as items and reposts as interactions. User attributes include name, gender, number of followers, etc. Item attributes include content and repost count.
- **Amazon** [14]: Amazon is an e-commerce dataset. Following [34], seven categories are combined: Automotive, Cell Phones and Accessories, Clothing, Shoes and Jewelry, Electronics, Grocery and Gourmet Food, Home and Kitchen, and Movies and TV. Attributes include item category, brand, features, description, user reviews, and others.

**Baseline details.** To ensure fairness, we use the same recommender backbone for ToP-Rec and the baselines. For reranking methods like MMR and DPP, we found that using 10 times the top-$k$ value as the candidate list can bring the performance close to its peak. For LLM4Rerank-A and LLM4Rerank-AD, we use twice the top-$k$ value due to their instability. We categorize the nine comparative methods used in this work into three types.

Heuristic methods:

- **Random**: Randomly generate a fixed number of user-item interactions and add them to the original history interaction set, then train the recommender.
- **MMR** [35]: A heuristic algorithm optimizing the trade-off between relevance and diversity, selecting items that are pertinent to the query and minimally redundant.
- **DPP** [36]: DPP is a parametric model for selecting a diverse subset from a larger pool of items. [36] accelerates DPP by proposing an algorithm that significantly speeds up the greedy MAP inference.

Conventional diversity-enhancing methods:

- **Box** [6]: A framework that enhances diversity and accuracy using box embedding to create hypercubes for users and items, with a similarity scoring function to measure their relationship.
- **LCD-UC** [6]: To better balance accuracy and diversity, LCD-UC adds an L-Step, C-Step, and D-Step design, along with an uncertainty masking mechanism, on top of box embedding.
- **CDM** [18]: A recommendation framework leveraging contrastive context encoder with attention mechanisms, distilling knowledge from MMR-based teacher output, and combining scores for diverse results.

LLM-based diversified recommenders:

- **LLM4Rerank-A** [20]: A reranking framework addressing accuracy, diversity, and fairness by abstracting requirements into interconnected nodes and dynamically adjusting aspect priorities through a Chain-of-Thought process. For LLM4Rerank-A, we set the LLM's focus solely on improving accuracy during reranking.
- **LLM4Rerank-AD** [20]: Similar to LLM4Rerank-A, the difference lies in setting the LLM's focus on improving both accuracy and diversity during reranking.
- **LLMRec-MMR** [24]: LLMRec uses LLMs to enhance recommendation relevance through three graph augmentation strategies: reinforcing user-item interactions, improving item attributes, and refining user profiling. MMR is then applied to balance diversity by reranking the list.

### A.4 More experiment results

**Empirical details of ToP.** During the construction of ToP, we prompt the LLM to generate a multi-level user preference tree (see § 3.2 and Appendix A.2). By instructing the LLM that each node represents a preference type and each edge denotes a finer-grained division, hierarchical nodes are ultimately generated. We provide additional empirical details of the tree of preferences below. As an illustration, key architectural specifications of ToP constructed on Twitter across runs include 133.6 leaf nodes, 248.4 total nodes, an average depth of 6.2, 53.3 average items per leaf, and an average branching factor of 2.16; to better showcase the structure of ToP, we then present a partial visualization of an actual ToP instance employed in the Twitter dataset, as shown in Figure 5.

**Empirical analysis of user selection.** The effectiveness of influential user selection in ToP-Rec is mainly influenced by three factors. First, the number of user groups determines group size, affecting user selection granularity. Second, the dimension of gradient reduction impacts the amount of retained information, and the step length $k$ affects the influence evaluation period. We analyze their impact on overall performance.

Figure 6(a) and (b) illustrate the changes in diversity and relevance on Twitter with different group numbers (maintaining a similar total number of augmented users) and different reduced dimensions. Increasing group size or dimensionality improves recall and category entropy, but with smaller marginal returns. This suggests that a small number of groups or dimensionality provides sufficient gains while keeping computational costs lower. Figure 6(c) visualizes the impact of step $k$, showing that smaller values (within 5) maintain low costs while yielding better performance. Larger step sizes degrade performance, likely due to the extended training history losing dynamic user influence.

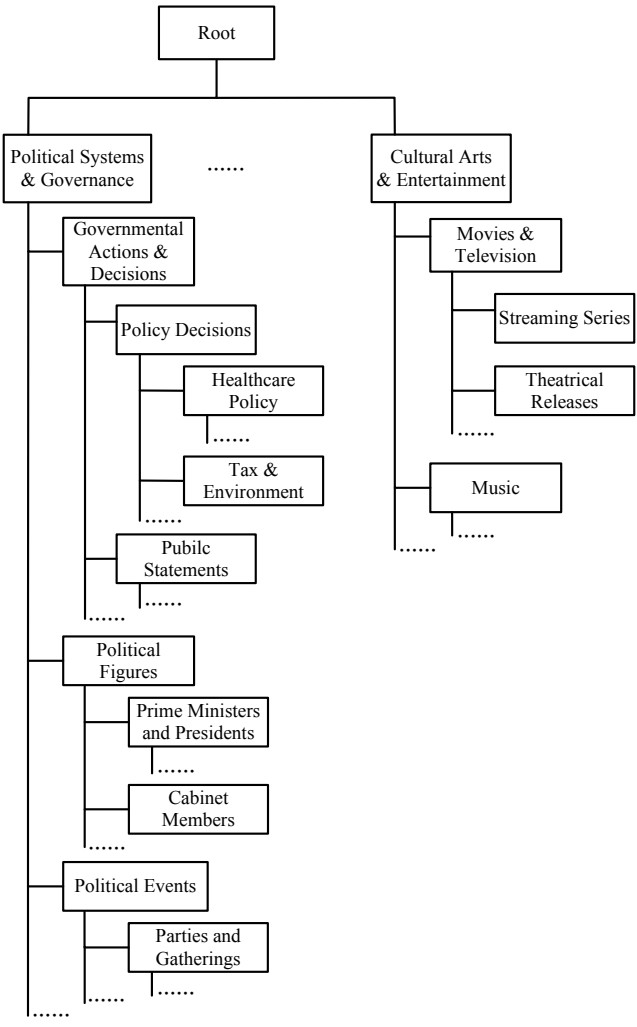

Figure 5: A partial visualization of the constructed ToP.

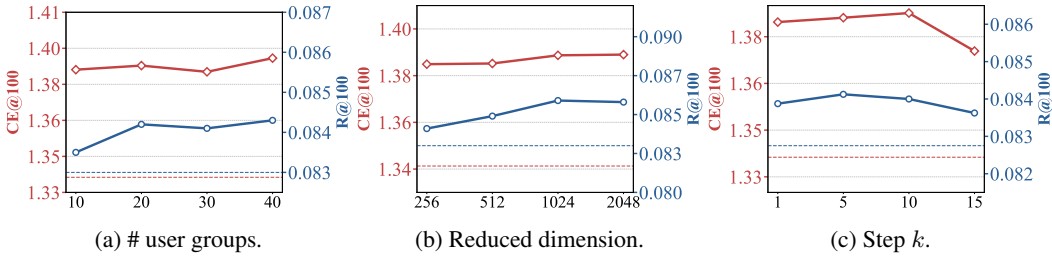

(a) # user groups.     (b) Reduced dimension.     (c) Step $k$.

Figure 6: Impact of (a) number of user groups, (b) reduced dimension, and (c) step $k$ on the performance of ToP-Rec. Dashed lines represent the performance of the backbone recommender.

**Performance evaluation on more recommender backbones.** We further evaluate the performance of ToP-Rec on two widely used recommenders, namely MF [32] and NGCF [78]. We set the hidden size to 32 and utilize the Adam optimizer with a learning rate of 5e-3, keeping other settings the same as LightGCN. We compare ToP-Rec with representative baselines, all reporting relatively balanced results for diversity and relevance. Tables 4 and 5 show the performance comparison on Twitter, using MF and NGCF as backbones, respectively. $^*$ denotes the backbone model, and $^+/^-$ indicates performance changes. The optimal performance is in bold, and the suboptimal is underlined. The

Table 4: Performance comparison using MF as recommender backbone.

| | R@50 | R@100 | CE@50 | CE@100 |
|---|---|---|---|---|
| MF[*] | 0.0329 | 0.0501 | 1.2249 | 1.3189 |
| MMR | 0.0319[−] | 0.0485[−] | 1.2596[+] | 1.3533[+] |
| LCD-UC | 0.0327[−] | 0.0503[+] | 1.2558[+] | 1.3504[+] |
| LLM4Re-AD | 0.0323[−] | 0.0498[−] | 1.2620[+] | 1.3444[+] |
| ToP-Rec | **0.0345**[+] | **0.0510**[+] | **1.2980**[+] | **1.3813**[+] |

Table 5: Performance comparison using NGCF as recommender backbone.

| | R@50 | R@100 | CE@50 | CE@100 |
|---|---|---|---|---|
| NGCF[*] | 0.0468 | 0.0748 | 1.2478 | 1.2977 |
| MMR | 0.0435[−] | 0.0708[−] | 1.2800[+] | 1.3261[+] |
| LCD-UC | 0.0466[−] | 0.0733[−] | 1.2897[+] | 1.3365[+] |
| LLM4Re-AD | 0.0467[−] | 0.0752[+] | 1.2722[+] | 1.3189[+] |
| ToP-Rec | **0.0473**[+] | **0.0763**[+] | **1.3630**[+] | **1.4028**[+] |

results show that ToP-Rec achieves optimal performance in both cases, further demonstrating its generalizability.

**Standard deviations.** Table 6 presents the standard deviations for the recall and category entropy shown in Table 1.

**Relevance-diversity trade-off.** We provide a trade-off analysis on the Weibo and Amazon datasets, as shown in Figure 7. Our approach consistently achieves the best trade-off, demonstrating its outstanding performance.

**Hyperparameter analysis.** We first examine the impact of the number of selected leaf nodes per user on ToP-Rec's performance. Varying the number of selected leaves among $\{4, 5, 6, 7\}$, the results on the Twitter dataset in Figure 8(a) show that as leaf nodes increase, diversity improves, while relevance rises initially and then slightly declines. This is likely due to more leaf nodes providing ToP-Rec with greater capacity to explore unobserved preferences. However, as the number of leaf nodes increases further, the risk of introducing noise also rises. Nevertheless, due to our systematic preference reasoning design, ToP-Rec's performance remains robust, showing substantial improvement.

Then, we evaluate the impact of the structure of ToP by adjusting prompts to modify requirements for tree depth and branching number. Figure 8(b) and (c) report ToP-Rec's performance across different runs, with variations in tree depth and average branching factor. We find that shallow structures constrain preference resolution, while excessive depth also degrades performance, potentially attributed to semantic fragmentation. Following a similar logic, the average branching factor exhibits a comparable trend: relatively fewer or more branches both lead to suboptimal performance.

**Stability analysis of ToP-Rec under key variations.** We further investigate the stability of ToP-Rec across variations in the applied LLMs, prompt strategies, and initial sampled item counts. Specifically, we integrate ToP-Rec with two more LLMs of distinct parameter scales and architectural designs, including Qwen2.5-7B-Instruct [37] and Llama3-70B [79]. Stability is evaluated along three key dimensions: the number of leaf nodes in the constructed ToP, the depth of ToP, and the final recommendation performance. Table 7 presents the results on the Twitter dataset. Our findings demonstrate that ToP-Rec maintains robust stability across different LLMs.

Next, we evaluate the impact of distinct prompt strategies on ToP's structural stability and the final recommendation performance. We design two variants: a simplified prompt (with specific task-related instructions removed) and an enhanced prompt (incorporating more detailed task guidance). Corresponding results presented in Table 8 reveal that the enhanced prompt leads to a modest increase

Table 6: Standard deviation.

| | Twitter | | | | Weibo | | | | Amazon | | | |
|---|---|---|---|---|---|---|---|---|---|---|---|---|
| | R@50 | R@100 | CE@50 | CE@100 | R@50 | R@100 | CE@50 | CE@100 | R@50 | R@100 | CE@50 | CE@100 |
| LightGCN[*] | 0.0008 | 0.0007 | 0.0030 | 0.0038 | 0.0002 | 0.0006 | 0.0036 | 0.0026 | 0.0007 | 0.0006 | 0.0071 | 0.0057 |
| Random | 0.0014 | 0.0024 | 0.0131 | 0.0145 | 0.0003 | 0.0005 | 0.0039 | 0.0065 | 0.0015 | 0.0024 | 0.0108 | 0.0121 |
| MMR | 0.0004 | 0.0004 | 0.0027 | 0.0021 | 0.0002 | 0.0002 | 0.0084 | 0.0076 | 0.0014 | 0.0014 | 0.0030 | 0.0059 |
| DPP | 0.0017 | 0.0015 | 0.0108 | 0.0073 | 0.0006 | 0.0009 | 0.0054 | 0.0073 | 0.0006 | 0.0007 | 0.0053 | 0.0050 |
| CDM | 0.0012 | 0.0008 | 0.0046 | 0.0051 | 0.0002 | 0.0003 | 0.0057 | 0.0055 | 0.0021 | 0.0019 | 0.0161 | 0.0162 |
| Box | 0.0018 | 0.0028 | 0.0058 | 0.0044 | 0.0028 | 0.0016 | 0.0078 | 0.0121 | 0.0023 | 0.0020 | 0.0072 | 0.0089 |
| LCD-UC | 0.0016 | 0.0032 | 0.0093 | 0.0085 | 0.0016 | 0.0024 | 0.0081 | 0.0039 | 0.0021 | 0.0018 | 0.0082 | 0.0078 |
| LLMRec-MMR | 0.0011 | 0.0012 | 0.0044 | 0.0057 | 0.0006 | 0.0008 | 0.0062 | 0.0050 | 0.0017 | 0.0019 | 0.0101 | 0.0092 |
| LLM4Re-A | 0.0008 | 0.0010 | 0.0098 | 0.0083 | 0.0003 | 0.0008 | 0.0055 | 0.0079 | 0.0013 | 0.0015 | 0.0089 | 0.0091 |
| LLM4Re-AD | 0.0010 | 0.0012 | 0.0161 | 0.0147 | 0.0004 | 0.0005 | 0.0096 | 0.0065 | 0.0025 | 0.0028 | 0.0156 | 0.0114 |
| ToP-Rec | 0.0012 | 0.0013 | 0.0127 | 0.0141 | 0.0006 | 0.0008 | 0.0066 | 0.0059 | 0.0010 | 0.0006 | 0.0142 | 0.0136 |

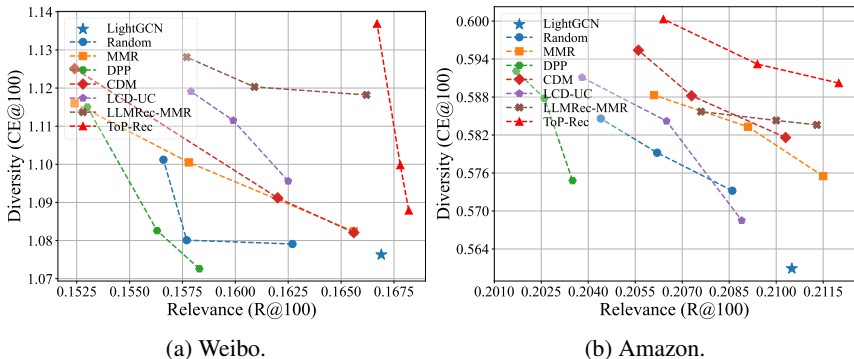

(a) Weibo.  (b) Amazon.

Figure 7: Diversity-relevance trade-off comparison. The upper-right represents the ideal.

in the number of leaf nodes within the ToP structure. Notably, regardless of the prompt variant employed, ToP-Rec consistently outperforms all baselines, demonstrating the framework's robustness to variations in prompt design.

Finally, we assess the impact of varying initial sampled item counts on the stability of ToP-Rec. Consistent with prior analyses, results are presented in Table 9. ToP-Rec maintains robust stability even with a reduced initial sampling ratio, exhibiting no significant performance degradation and continuing to demonstrate superiority over baselines.

**Comparison with generative LLM-based recommendation.** We also provide a comparison with the state-of-the-art LLM-based generative diversified recommender, DLCRec [21]. Since DLCRec generates recommendations by predicting item names rather than IDs, it struggles with scenarios like Twitter or Weibo, where items are posts with long text and lack explicit names. Therefore, we conducted evaluations on Amazon, where items are products with brief names. We followed DLCRec's paradigm for LLM fine-tuning and generated 10 items per user during inference, calculating recall@10 and category-entropy@10. The results are shown in Table 10, indicating a significant improvement for our approach. We empirically found that DLCRec carries a high risk of generating homogeneous items, which can be attributed to its limited ability to fully capture users' latent preferences, leading to poorer performance. Furthermore, due to the instability of LLM generation, DLCRec performs poorly under our original settings, where metrics are evaluated at Top 50 and 100.

## A.5 Limitation and future work

While ToP-Rec shows promise in diversifying recommendations, two limitations need attention. First, the quality of user and item textual attributes affects its effectiveness. In cases of minimal or low-quality text, such as platforms with primarily video or image content, the current design may perform less effectively. Second, ToP-Rec does not focus on enhancing recommendation novelty. This limitation arises from prioritizing diversity in preference-aligned item matching without

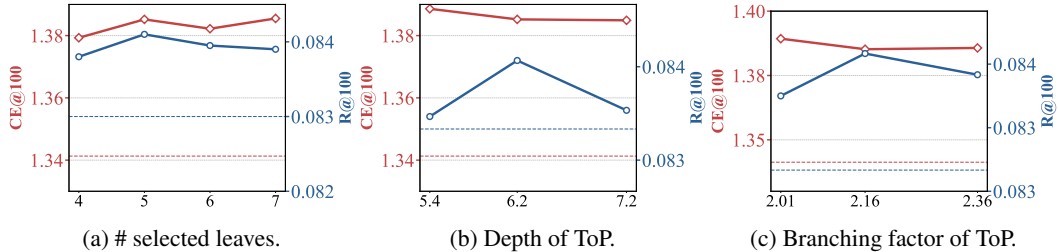

| (a) # selected leaves. | (b) Depth of ToP. | (c) Branching factor of ToP. |

Figure 8: Hyperparameter analysis of (a) number of selected leaves, (b) depth of ToP, and (c) branching factor of ToP. We use dashed lines to represent the performance of the backbone recommender.

Table 7: Stability analysis across different LLMs.

|  | # leaf nodes | Depth | R@50 | R@100 | CE@50 | CE@100 |
|---|---|---|---|---|---|---|
| Llama3-70B | 137.4 | 6.8 | 0.0584 | 0.0840 | **1.3367** | **1.3940** |
| Qwen2.5-7B-Instruct | 138.2 | 7.0 | 0.0579 | 0.0836 | 1.3291 | 1.3857 |
| Qwen2.5-32B-Instruct (Ours) | 133.6 | 6.2 | **0.0586** | **0.0841** | 1.3275 | 1.3852 |

explicitly quantifying each item's potential to surprise users. Future work will focus on integrating novelty-aware objectives, further enhancing the user experience.

Table 8: Stability analysis across different prompt strategies.

|  | # leaf nodes | Depth | R@50 | R@100 | CE@50 | CE@100 |
|---|---|---|---|---|---|---|
| Simple prompt | 129.4 | 6.6 | 0.0583 | **0.0853** | **1.3329** | 1.3821 |
| Complex prompt | 147.8 | 6.0 | 0.0573 | 0.0846 | 1.3208 | **1.3853** |
| Ours | 133.6 | 6.2 | **0.0586** | 0.0841 | 1.3275 | 1.3852 |

Table 9: Stability analysis across different initial sampling ratios.

|  | # leaf nodes | Depth | R@50 | R@100 | CE@50 | CE@100 |
|---|---|---|---|---|---|---|
| 3% sampling | 137.4 | 6.6 | 0.0575 | 0.0840 | **1.3284** | 1.3845 |
| 6% sampling | 126.8 | 6.8 | 0.0583 | 0.0840 | 1.3273 | 1.3830 |
| 9% sampling (Ours) | 133.6 | 6.2 | **0.0586** | **0.0841** | 1.3275 | **1.3852** |

Table 10: Performance comparison with DLCRec.

|  | R@10 | CE@10 |
|---|---|---|
| DLCRec | 0.0021 | 0.4714 |
| ToP-Rec | **0.0119** | **0.9639** |

