# OpenReview forum: "Tree of Preferences for Diversified Recommendation"
_NeurIPS.cc/2025/Conference — NeurIPS 2025 poster_

### Official Review · Reviewer_2PtT · 2025-06-30

**Clarity:** 2
**Significance:** 3
**Originality:** 3
**Rating:** 4
**Confidence:** 4

**Summary:**

This paper proposes a method for improving diversified recommendation by addressing potential data biases, such as exposure and selection bias, which may obscure users' underexplored preferences in observed interactions. The authors introduce the Tree of Preference, a hierarchical structure designed to guide LLMs in reasoning about user behavior from coarse to fine granularity. Based on the inferred preferences, the approach generates synthetic interactions to supplement training data and enhance diversity in recommendations. A dynamic user selection mechanism is also employed to improve training efficiency. Experimental results show that the method achieves competitive performance in terms of both diversity and relevance, with inference latency remaining within a reasonable range.

**Questions:**

See limitations.

**Ethical Concerns:**

["NO or VERY MINOR ethics concerns only"]

**Final Justification:**

Based on the quality of the manuscript and the rebuttal, most of my concerns have been addressed. Authors should combine with rebuttal contents to enhance the manuscript. I recommend for 4: Borderline accept.

**Limitations:**

yes

**Quality:**

3

**Strengths And Weaknesses:**

Strengths：
1. The authors propose a tree-based structure to guide LLMs in reasoning about users’ latent preferences, which introduces a degree of novelty to the modeling approach.
2. The figures and tables are clearly presented and easy to interpret, which aids understanding of the methodology and results.

Weaknesses：
1. While the paper acknowledges the presence of exposure and selection bias in user interaction data and aims to mitigate these biases through supplementary knowledge, it still relies entirely on offline datasets for training and evaluation. As a result, the test and validation sets may suffer from the same biases, making it difficult to assess whether the model truly captures users’ latent interests.
2. The proposed method is highly heuristic in nature and heavily relies on a pretrained LLM without any task-specific fine-tuning, which may limit its adaptability and effectiveness.
3. The approach depends critically on the constructed preference tree, yet the paper does not provide sufficient detail on how the tree is built, such as the number of levels, how nodes are generated, how many leaf nodes exist, or how many items each leaf node corresponds to. This lack of clarity significantly hinders reproducibility.
4. The hyperparameter analysis does not include any discussion related to the construction of the preference tree, making it difficult to understand how different tree structures may affect the model’s performance.
5. The proposed method assumes the availability of rich user profiles or user-related textual information. However, in many practical recommendation scenarios, such data may be limited or unavailable due to privacy constraints, which could restrict the applicability of the method.

---

> ### Author Rebuttal · Authors · 2025-07-31
>
> > W1: Test and validation sets may suffer from the same biases, making it difficult to assess whether the model truly captures users’ latent interests.
>
> We sincerely appreciate your insightful comment. We acknowledge that conventional offline datasets may carry biases in test and validation sets. However, online validation typically requires randomized interventions (e.g., A/B testing), which present practical challenges such as high costs and reproducibility issues.
>
> To rigorously evaluate ToP-Rec under biased test assumptions, we follow established work in debiasing evaluation [1], which aims to conduct unbiased evaluation when offline datasets exhibit selection or exposure bias.  We focus on relevance evaluation under these biased assumptions, using two strategies: Inverse Propensity Scoring Recall ($ R_{IPS} $) [2] and ATOP [3]. $ R_{IPS} $ weights observations using propensity scores, providing an unbiased recall estimator, while ATOP uses item set ranking, also proven to be unbiased for relevance. We report ToP-Rec's performance,  compared with both the backbone recommender and comparable baselines. The results show that ToP-Rec retains significant advantages under the assumption that test and validation sets may contain inherent biases.
>
> |   | $\text{R}_\text{IPS}$@50  | $\text{R}_\text{IPS}$@100 | ATOP |
> |----|--------|-----------|-------|
> |LightGCN$^*$  |0.0138|0.0188|0.4936|
> |LCD-UC|0.0122|0.0178|0.4915|
> |LLM4Re-A |0.0134|0.0181|0.4905|
> |Ours|**0.0161**|**0.0221**|**0.5008**|
>
> Since the calculation of category entropy depends solely on the distribution of categories in an item list and is not influenced by unobserved user-item interactions, our diversity evaluation remains unaffected by bias.
>
> > W2: The proposed method is highly heuristic in nature and heavily relies on a pretrained LLM without any task-specific fine-tuning, which may limit its adaptability and effectiveness.
>
> Thank you for raising this important point, which inspired us to clarify Top-Rec's design.
>
> Firstly, we would like to emphasize that the key contribution of Top-Rec lies in its general framework rather than the incidental use of LLMs. The tree of preference serves as a structured vehicle for uncovering underexplored user preferences. Our experiments demonstrate that Top-Rec is compatible with a variety of LLMs, including smaller-scale models, which maintain robust performance (Please refer to our response to Q1 from Reviewer K1gm and W1 from Reviewer F6jv). This highlights the framework’s advantages.
>
> Secondly, we utilize pretrained LLMs because Top-Rec's fundamental objective is to harness their rich world knowledge for latent preference discovery, not as specialized solvers for complex tasks (e.g., mathematical reasoning or code generation). Since pretrained LLMs inherently satisfy this requirement, we find them well-suited to our design. Empirical evidence consistently confirms Top-Rec's effectiveness.
>
> Following your insightful suggestion, we further evaluated Top-Rec with fine-tuning. Specifically, we fine-tuned the LLM for user preference inference, which is a key step in the Top-Rec process. Using Twitter training data, we constructed a fine-tuning dataset where the LLM is instructed to infer preferences from users' historical interactions and the constructed Tree-of-Preference. We applied LoRA to fine-tune the LLM used in the paper and compared its performance with the original Top-Rec. The results showed that the fine-tuned version did not lead to significant improvements, confirming that pretrained LLMs are sufficiently effective while offering greater flexibility for diverse scenarios.
>
> |   | R@50  | R@100 | CE@50 | CE@100 |
> |----|--------|-----------|-------|-------|
> |Fine-tuned|0.0583|0.0829|1.3252|1.3841|
> |Ours|**0.0586**|**0.0841**|**1.3275**|**1.3852**|
>
>
> > W3: The paper does not provide sufficient detail on how the tree is built.
>
> We sincerely appreciate your valuable comment. Regarding how nodes are generated, during the construction of the ToP, we prompt the LLM to generate a multi-level tree structure for user preferences. By instructing the LLM that each node represents a type of user preference and each edge represents a finer preference division, the hierarchical nodes are ultimately generated by the LLM. Please refer to our example prompt in Appendix A.2 and the actual ToP presented in our response to Reviewer K1gm, W2.
>
> Below, we provide key architectural specifications of the tree of preference constructed on Twitter across different runs, including the number of leaf nodes, total nodes, depth, average items assigned per leaf, and average branching factor. To enhance reproducibility, we will revise the paper to include detailed quantitative information about the tree structure, along with a visualization of the ToP.
>
> | # leaf nodes  | # nodes | Depth  | # items per leaf  | Branch factor |
> |----|--------|-----------|-------|-------|
> |133.6|248.4|6.2|53.3|2.16|
>
> > W4: Hyperparameter analysis does not include discussions related to the construction of the preference tree.
>
> Thank you for reminding us about this issue. We conduct evaluations to assess how the construction of the tree of preference influences performance.
>
> First, we examine how the number of sampled items provided to the LLM for ToP construction affects performance. As shown in the following table, we observe that Top-Rec achieves strong performance with a relatively small sampling ratio, which can be attributed to the fact that our k-means-based item sampling strategy helps maintain the representativeness of the samples.
>
>
> | Sampling ratio  | R@50  | R@100 | CE@50 | CE@100 |
> |----|--------|-----------|-------|-------|
> |3%|0.0575|0.0840|**1.3284**|1.3845|
> |6%|0.0583|0.0840|1.3273|1.3830|
> |9%|**0.0586**|**0.0841**|1.3275|**1.3852**|
>
> Second, we evaluate the impact of the ToP structure by adjusting the prompt to modify the requirements for tree depth and branching number. In the two tables below, we report Top-Rec's performance on Twitter with varying tree depth and average branching factor across different runs. We find that shallow structures limit preference resolution, while excessive depth also restricts performance, potentially due to semantic fragmentation. Following a similar logic, the average branching factor exhibits a comparable trend, where relatively fewer or more branches lead to suboptimal performance.
> We apologize for not clearly articulating this connection in the paper and will include a dedicated discussion of these findings in the revised version.
>
> | Depth  | R@50  | R@100 | CE@50 | CE@100 |
> |----|--------|-----------|-------|-------|
> |5.4|0.0565|0.0832|**1.3324**|**1.3886**|
> |6.2|**0.0586**|**0.0841**|1.3275|1.3852|
> |7.2|0.0572|0.0833|1.3295|1.3849|
>
>
> | Branch factor  | R@50  | R@100 | CE@50 | CE@100 |
> |----|--------|-----------|-------|-------|
> |2.01 |0.0571|0.0837|1.3269|**1.3893**|
> |2.16|**0.0586**|**0.0841**|1.3275|1.3852|
> |2.36|0.0584|0.0839|**1.3338**|1.3857|
>
>
> > W5: Availability of rich user profiles.
>
> Thank you for your thoughtful comment. Top-Rec aligns with existing LLM-based recommendation methods [4,5] in terms of its reliance on basic user profile information, without requiring more detailed user data. This basic information is typically easy to obtain and leverage in industrial recommendation scenarios [6,7,8].
>
> More importantly, following the approach in [4], we enhance user data by incorporating item information from the user’s historical interactions. These items serve as an additional, rich source of user information. We collect the textual features of items the user interacted with and provide them alongside the basic user profile information to the LLM for summarization. By focusing on user interests and filtering out noise, we obtain enhanced user textual data.
>
> To evaluate the applicability of our method in scenarios where user profiles are missing, we conducted additional experiments. Specifically, we masked the user profiles and used only the LLM-generated summarization from the historically interacted items as user data. The table below presents the results on Twitter, showing that our method still outperforms strong baselines, demonstrating its robustness.
>
> |   | R@50  | R@100 | CE@50 | CE@100 |
> |----|--------|-----------|-------|-------|
> |LightGCN$^*$  |0.0567|0.0830|1.2841|1.3413|
> |LCD-UC|0.0517|0.0768|1.3154|1.3784|
> |LLM4Re-A|0.0562|0.0827|1.2855|1.3424|
> |w/o profile|$\underline{0.0584}$|$\underline{0.0839}$|$\underline{1.3258}$|**1.3895**|
> |Ours|**0.0586**|**0.0841**|**1.3275**|$\underline{1.3852}$|
>
>
>
> [1] Bias and Debias in Recommender System: A Survey and Future Directions
>
> [2] Unbiased offline recommender evaluation for missing-not-at-random implicit feedback
>
> [3] Training and Testing of Recommender Systems on Data Missing Not at Random
>
> [4] LLMRec: Large Language Models with Graph Augmentation for Recommendation
>
> [5] LLM4Rerank: LLM-based Auto-Reranking Framework for Recommendations
>
> [6] Large Language Model as Universal Retriever in Industrial-Scale Recommender System
>
> [7] Towards Open-World Recommendation with Knowledge Augmentation from Large Language Models
>
> [8] On Practical Diversified Recommendation with Controllable Category Diversity Framework

---

> > ### Author Response · Authors · 2025-08-04
> >
> > Dear Reviewer,
> >
> > Thank you for your review and comments. We hope that our rebuttal and additional evaluations have addressed your concerns. We would greatly appreciate any feedback on any existing or new points we may not have covered, and we would be glad to address or discuss them further.
> >
> > Best regards,
> > Authors

---

> > > ### Comment · Reviewer_2PtT · 2025-08-04
> > >
> > > Thanks for your reply. Most of my concerns have been addressed. Therefore, I am happy to raise my score.

---

> > > > ### Author Response · Authors · 2025-08-05
> > > >
> > > > We are happy to hear that our response was helpful.​​ Thank you for your ​​​​prompt feedback and ​​recognition. We will ​​certainly​​ incorporate your valuable suggestions into the revised manuscript.

---

### Official Review · Reviewer_5Xsf · 2025-07-02

**Clarity:** 3
**Significance:** 3
**Originality:** 3
**Rating:** 5
**Confidence:** 4

**Summary:**

The authors present a method (Top-Rec) for using LLMs to promote diversity while maintaining relevancy in recommendations. Users and items are assumed to have textual descriptions available. An LLM is asked to constructed a detailed, multi-layer tree of preferences (ToP) organizing a representative sampled set of items. Given the ToP and a list of items a user likes, the LLM is then asked to generate a list of leaf nodes from the ToP representing the user interests, where implicit, latent preference leaf nodes are extracted as well as the more obvious ones. Synthetic data is then generated from items which fall under the implicit leaf nodes and that synthetic data is added to the real data for the sake of training a recommendation system. For the sake of of efficiency, particularly influential users are identified using gradient alignment methods and the generation of synthetic data focuses on those influential users. Highly positive results are obtained on Twitter, Weibo and Amazon datasets in terms of both relevance and diversity of recommendations, with Top-Rec beating all competitors in 11 out of 12 comparisons. Ablation studies are presented to confirm the significance of both diversity and relevance criteria as well as the utility of user influence measures.

**Questions:**

My questions have mostly already been outlined in the strengths weaknesses section, but to reiterate:

1) What can you tell me to assure me that there is no test set contamination which would help Qwen "cheat" in subtle statistical ways when helping to generate synthetic data for Amazon, Weibo, etc?

2) Is the tree of preferences a single, global tree or a per-user tree? Can you edit the paper to make this clearer?

3) Does the method require significant user-profile text or can you substitute text from items the user likes?

4) Can you tell me more about how prompt-length limits interact with constraints on how many sample items can be supplied to the LLM and how long the item descriptions can be? Can you assure me that this won't limit the utility of the method in other contexts?

**Ethical Concerns:**

["NO or VERY MINOR ethics concerns only"]

**Final Justification:**

The authors resolved my confusion regarding (1) whether or not the tree is a single tree, (2) prompt-length constraints and (3) whether the method relies on substantial user profile text.

I have been reviewing for NeurIPs for many years, and in my experience, it is rare to see all reviewers rate a paper as 4 or above. In light of this uniformity of agreement, I decided to raise my score to a 5.

**Limitations:**

Adequately addressed.

**Paper Formatting Concerns:**

No concerns.

**Quality:**

3

**Strengths And Weaknesses:**

The work is original to the best of my knowledge.

The experimental results are quite impressive. If indeed the experimental results are indicative of the improvements TopRec could generate in real-life settings, the work should have substantial significance.

One concern I have is the possibility of some sort of test set contamination. Qwen2.5-32B-Instruct was created after all 3 datasets were created. Is it possible that some of the crawled text that Qwen absorbed somehow lets it cheat in creating the tree of preferences, assigning the items to leaves, etc? My understanding is that test set leakage can arise in subtle and surprising ways with LLMs.

I also have a clarity concern which may be my fault, but maybe not. I'm slightly embarrassed to ask this, but if I have to ask this, I think it's fair to wonder whether the paper could be clearer on this point,  Is the tree of preferences created one-time, globally, across all users, or is there a per-user tree of preference? Maybe this should be clear to me, but I went over the paper several times and still wasn't totally sure what the answer is here. Again, maybe the fault is mine for not reading carefully enough, but I've won top-reviewer NeurIPs awards multiple times, so if it wasn't clear to me, then I think the paper could probably be clearer on this point.

I am curious about the typical limits of prompt length for LLMs and whether that does or could pose constraints or difficulties for this approach when supplying the LLM with a list of items. Clearly, the results are positive, so this constraint wasn't a big deal here, but are we sure that the sampling will be able to overcome the prompt length constraints in the future? I'd be interested in some simple stats like e.g. Qwen can take prompts of max length X characters and if an item description has an average length of Y characters then that means we can put Z items in a prompt and this should usually be enough because <insert reasons here>.

I am also curious about the cosine similarity method on line 210 and its reliance on A_u. More generally, I am curious about the importance of the assumption of the existence of user profile text A_u. Might it not be better (and more widely applicable) instead of using text from A_u to use a summary of text from items the user likes, i.e. from their real interaction history? I want to make sure I understand A_u.  A_u is (for instance) the profile text from Figure 1 (Bio: peaks unclimbed, etc) , correct? Does your method need such a profile? I could imagine many applications where no such profile (or a profile with minimal, useless text) exists, but we still have the reposts and likes (Summer escape), etc? Is there any reason you can't use a sample of the reporsts and likes if the user profile is not available?

Some suggested rewrites and typos;

Line 32 refers to the interaction data on line 30 as 'implicit feedback', but ratings are explicit, not implicit

Line 50 a diversified recommendation -> a diversified set of recommendations.

Line 8 appendix Frist -> First

Appendix, page 2, figure 4: Notation, not Notaion

*** Update after rebuttal ****

In light of the clarifications supplied by the authors as well as the uniformly positive ratings from other reviewers (which is rare to see), I have chosen to increase my score to a '5'.

---

> ### Author Rebuttal · Authors · 2025-07-31
>
> >Q1: Test set contamination which would help Qwen "cheat".
>
> Thank you for raising this valuable concern. Research on test set contamination in LLM-based recommender systems is indeed challenging. We address this by systematically analyzing risks at each stage of Top-Rec.
>
> First, during the construction of the tree of preference and assignment of items to leaves, Top-Rec exclusively utilizes item-side textual information. This process is orthogonal to user-item interaction data in the test set. We believe test set contamination is highly unlikely at this stage.
>
> Second, in the synthetic data generation phase, the LLM indirectly participates in user preference inference. While LLMs possess the capability to memorize the pretraining texts, this memorization ability is unlikely to extend to user-item relations, as such interactions are typically not stored as plain text. Consequently, the feasibility of the LLM cheating in preference inference by memorizing specific interactions remains low.
>
> To quantitatively evaluate this risk, we adopted the LLM pretraining data detection method [1], which identifies whether arbitrary data points belong to the LLM’s training data. Specifically, to investigate potential leakage of test set user-item interaction, we constructed two text corpora (10,000 samples each) describing user-item relationships (e.g., "User A with profile B liked Tweet C with content D"). One corpus contained ground-truth relationships from the test set, while the other contained randomly paired false relationships serving as a control group. We applied a state-of-the-art detector MIN-K% [1] to detect the likelihood of these relationships belonging to Qwen2.5-32b-Instruct’s training data. Results from MIN-K% showed that the proportion of test set relationships identified as **non-training data (99.7%)**  was nearly identical to that of randomly generated false relationships (99.5%), indicating minimal test set leakage risk. We further conducted significance testing on the perplexity distributions of both text groups, which reflect the model’s predictive confidence.  We applied two statistical tests: the t-test and the Kolmogorov–Smirnov  test. Both yielded **p-values above 0.9**, showing no significant statistical difference between the distributions of the perplexity scores. This confirms that Qwen’s perplexity for true and false interactions is statistically indistinguishable, providing additional evidence against test set contamination.
>
> >Q2: Is the tree of preferences a single, global tree or a per-user tree?
>
> We sincerely apologize for any confusion caused. We would like to clarify that the tree of preference is constructed as a **global, one-time structure shared across all users**. This design is based on our understanding that, while individual preferences may vary, all users ultimately navigate the same underlying preference space defined by item characteristics. We have revised the paper to clarify this point.
>
>
> >Q3: Does the method require significant user-profile text or can you substitute text from items the user likes?
>
> Thank you for raising this important point.
>
> Firstly, the user profile refers to basic registration information, as shown in Figure 1 (e.g., age, bio). Similar to prior work [2,3], our method incorporates such a profile, but does not rely on a large amount of user-profile text. The profile is not the only source of user information. As you rightly pointed out, we also utilize the text attributes from items the user has interacted with, which provide valuable insights into their preferences. These interacted items serve as another crucial source of user information. Ultimately, we follow [2] to provide both the text attributes from user profile and the items they interacted with to the LLM. By focusing on the user's interests and filtering out noise, we obtain an enhanced user summary from the LLM as the final textual feature.
>
> When the user profile is unavailable, Top-Rec can rely on the text attributes from the interacted items as a substitute to generate the user's textual feature via LLM. To evaluate performance in such scenarios, we masked the original user profile and used only summaries from the interacted items as users' text features, then evaluated Top-Rec on this dataset. The table below presents the results on Twitter, which show that our method still outperforms the comparative baselines.
> |   | R@50  | R@100 | CE@50 | CE@100 |
> |----|--------|-----------|-------|-------|
> |LightGCN$^*$  |0.0567|0.0830|1.2841|1.3413|
> |LCD-UC|0.0517|0.0768|1.3154|1.3784|
> |LLM4Re-A|0.0562|0.0827|1.2855|1.3424|
> |w/o profile|$\underline{0.0584}$|$\underline{0.0839}$|$\underline{1.3258}$|**1.3895**|
> |Ours|**0.0586**|**0.0841**|**1.3275**|$\underline{1.3852}$|
>
> >Q4: How prompt-length limits interact with constraints on how many sample items can be supplied to the LLM and how long the item descriptions can be? And the utility of the method in other contexts.
>
> Thank you for raising this important question. We acknowledge that such constraints warrant consideration, but they do not significantly impact the construction of the tree of preference. This is because ToP relies not on the absolute size of the entire item set, but on the semantic breadth of a sampled subset. Crucially, our approach focuses on identifying a representative subset that captures the underlying preference space of the full item collection.
>
> To achieve this, we designed a k-means clustering-based sampling method (details in Appendix A.2). By clustering items and sampling proportionally from each cluster, we obtain a compact yet representative subset for LLM processing. As an empirical example, Qwen2.5 supports up to 131,072 tokens of context, and in the Twitter dataset, each item averages 27.6 tokens. This results in a maximum theoretical capacity of approximately 4749.0 items per prompt. In practice, our sampled subsets consist of around 650 items (13.7% of the capacity), demonstrating substantial operational headroom.
>
> We also investigate the impact of the number of sampled items on Top-Rec's performance. As illustrated in the table below, we find that Top-Rec performs well even with a lower sampling ratio.
>
> | Sampling ratio  | R@50  | R@100 | CE@50 | CE@100 |
> |----|--------|-----------|-------|-------|
> |3%|0.0575|0.0840|**1.3284**|1.3845|
> |6%|0.0583|0.0840|1.3273|1.3830|
> |9%|**0.0586**|**0.0841**|1.3275|**1.3852**|
>
> In other contexts, we expect the utility of ToP to remain robust against prompt length constraints. Our sampling method intrinsically mitigates such limitations, and the trend toward longer LLM contexts provides additional flexibility. In extreme cases, potential solutions include partitioned ToP construction based on auxiliary metadata (e.g., item categories).
>
> > Suggested rewrites and typos.
>
> We greatly appreciate your valuable feedback. As you suggested, we have carefully reviewed the paper, addressed the imprecise writing, and corrected the typos. Additionally, we have enhanced the paper to ensure a clearer and more rigorous presentation.
>
>
> [1] Detecting Pretraining Data from Large Language Models
>
> [2] LLMRec: Large Language Models with Graph Augmentation for Recommendation
>
> [3] LLM4Rerank: LLM-based Auto-Reranking Framework for Recommendations

---

> > ### Comment · Reviewer_5Xsf · 2025-08-02
> > **Thank you for the rebuttal**
> >
> > I have read and do appreciate the authors rebuttal, which touches on guardrails against test set contamination, use of profile text vs. item preference history and prompt length constraints. This response is very helpful. I'm a little surprised that 650 items is enough to generate a global tree, but I supposed the tree leaves don't have to be very fine-grained for the method to be useful.
> >
> > I will certain keep this rebuttal in mind as we moved into the final recommendation phase.
> >
> > If I have further questions for the authors, I will let them know.

---

> > > ### Author Response · Authors · 2025-08-03
> > >
> > > Thank you for your prompt feedback and thoughtful comments. We’re so pleased to hear you found our response helpful! If further questions arise, we’d be delighted to discuss them with you.

---

### Official Review · Reviewer_F6jv · 2025-07-03

**Clarity:** 3
**Significance:** 2
**Originality:** 2
**Rating:** 4
**Confidence:** 3

**Summary:**

This paper introduces ToP-Rec, a novel framework designed to improve diversity in recommender systems by addressing inherent data biases. The core contribution is the "Tree of Preferences" (ToP), a hierarchical item structure generated by a Large Language Model (LLM) to represent its world knowledge. The LLM uses this tree to reason about a user's latent interests and generate synthetic interaction data, specifically targeting underexplored preference areas. This augmented data is then used to train a conventional, lightweight recommender model. This data-centric approach separates the expensive LLM reasoning from the efficient online serving phase, with an additional strategy to select influential users for augmentation to manage computational costs.

**Questions:**

1. The method's validity depends on the stability of the generated Tree of Preferences. How sensitive is the ToP structure to variations in the prompt, the choice of LLM, or the initial item sampling? A quantitative analysis of the tree's structural variance across different runs is needed to establish the method's reliability.
2. Could you provide a full breakdown of the offline computational costs (e.g., in GPU-hours) and compare the proposed influential user selection method against simpler, standard active learning heuristics (e.g., uncertainty sampling)? This is needed to justify the method's complexity and assess its true cost-efficiency.

**Ethical Concerns:**

["NO or VERY MINOR ethics concerns only"]

**Limitations:**

yes in appendix.

**Quality:**

2

**Strengths And Weaknesses:**

Strengths:
This paper presents a highly original and significant approach to diversified recommendation by reframing it as a data bias problem solvable with LLM-generated data. The primary strength is the "Tree of Preferences" (ToP) methodology, an innovative application of tree-based reasoning that allows an LLM to infer a user's unobserved interests in a structured, coarse-to-fine manner. This moves beyond simply augmenting existing data and into the more ambitious realm of correcting systemic bias by filling knowledge gaps. The system design is also elegant and practical, using the powerful but costly LLM offline to generate debiased training data, which then informs a fast, scalable conventional model for real-time inference.

Weaknesses:
The work exhibits some areas for potential improvement, including the current lack of a comprehensive robustness analysis for the ToP generation across different LLM models or prompt variations, which could further solidify the reliability of its foundational artifact. Additionally, while inference latency is well-addressed, the paper could provide more detailed quantification of the substantial offline computational costs, and further elaborate on the justification for its complex gradient-based influential user selection method, perhaps with more direct comparisons to simpler active learning heuristics. Lastly, the choice of a simple linear combination objective function for synthetic item selection, analogous to MMR, might benefit from additional exploration or justification regarding its optimality in this crucial data generation step.

---

> ### Author Rebuttal · Authors · 2025-07-31
>
> >  W1 & Q1: Analysis for the ToP generation across different LLM models, prompt variations and initial item sampling.
>
> We appreciate your insightful comment. To improve the stability of the ToP, we employ several designs, including using a k-means sampling strategy during the ToP construction to select a subset of items that is sufficiently representative of user preferences, thereby reducing the impact of randomness. Additionally, we incorporate a refinement mechanism that enhances the balance of the ToP through merge and split operations (detailed in Appendix A.2).
>
> Following your suggestion, we further evaluate the stability of the ToP with different LLMs, different prompt strategies, and varying numbers of initial sampled items. First, we integrate ToP-Rec with LLMs with different parameter sizes and architectures, including Qwen2.5-7B-Instruct and Llama3-70B. We report the stability of the ToP in terms of the number of leaf nodes, depth, and final performance. The following table presents the evaluation on Twitter, in comparison with the backbone recommender and other comparative baselines. The results show that ToP is relatively stable across different LLMs, with the final performance consistently outperforming the baselines.
>
> |   | # leaf nodes| Depth  | R@50  | R@100 | CE@50 | CE@100 |
> |----|--------|--------|-----------|-------|-------|-------|
> |LightGCN$^*$  |-|-|0.0567|0.0830|1.2841|1.3413|
> |LCD-UC|-|-|0.0517|0.0768|1.3154|1.3784|
> |LLM4Re-A|-|-|0.0562|0.0827|1.2855|1.3424|
> |Llama3-70B|137.4|6.8|0.0584|0.0840|**1.3367**|**1.3940**|
> |Qwen2.5-7B-Instruct |138.2|7.0|0.0579|0.0836|1.3291|1.3857|
> |Qwen2.5-32B-Instruct(Ours)|133.6|6.2|**0.0586**|**0.0841**|1.3275|1.3852|
>
> Next, we apply different prompt strategies, including a simpler version where we remove some specific task instructions, and a more complex version where we provide more detailed task instructions. Similarly, we report the evaluation in the following table. The results show that the more complex prompt leads to a slight increase in the number of leaf nodes in the ToP, while all versions of ToP-Rec outperform the baselines.
>
> |   | # leaf nodes| Depth  | R@50  | R@100 | CE@50 | CE@100 |
> |----|--------|--------|-----------|-------|-------|-------|
> |LightGCN$^*$  |-|-|0.0567|0.0830|1.2841|1.3413|
> |LCD-UC|-|-|0.0517|0.0768|1.3154|1.3784|
> |LLM4Re-A|-|-|0.0562|0.0827|1.2855|1.3424|
> |Simple prompt|129.4|6.6|0.0583|**0.0853**|**1.3329**|1.3821|
> |Complex prompt|147.8|6.0|0.0573|0.0846|1.3208|**1.3853**|
> |Ours|133.6|6.2|**0.0586**|0.0841|1.3275|1.3852|
>
> Finally, we examine the impact of varying numbers of initial sampled items. Similar to the previous analysis, we report the evaluation in the table below. We observe that ToP remains relatively stable, even with a lower sampling ratio.
>
> |    |# leaf nodes| Depth  | R@50  | R@100 | CE@50 | CE@100 |
> |----|--------|-----------|-------|-------|-------|-------|
> |LightGCN$^*$  |-|-|0.0567|0.0830|1.2841|1.3413|
> |LCD-UC|-|-|0.0517|0.0768|1.3154|1.3784|
> |LLM4Re-A|-|-|0.0562|0.0827|1.2855|1.3424|
> |3% sampling|137.4|6.6|0.0575|0.0840|**1.3284**|1.3845|
> |6% sampling|126.8|6.8|0.0583|0.0840|1.3273|1.3830|
> |9% sampling(Ours)|133.6|6.2|**0.0586**|**0.0841**|1.3275|**1.3852**|
>
> >  W2 & Q2: Detailed quantification of offline computational costs and comparison to simpler active learning.
>
> Thank you for your valuable comment. We’ve provided a detailed breakdown of the main steps in training Top-Rec that contribute to the overall computational costs. For tasks performed on our local device, we provide the computational cost in GPU hours based on a single NVIDIA RTX 3090. For the steps executed via the LLM API, we compute the equivalent computational cost expressed in GPU hours on a single NVIDIA A100. The computational cost analysis for Top-Rec on the Twitter dataset is summarized in the table below.
>
> |  Task | GPU hour  | GPU type |
> |----|--------|-----------|
> |Constructing ToP  | 1.14 | A100 |
> |Matching items with ToP| 0.67 | A100 |
> |Training recommender backbone| 0.16| RTX3090 |
> |Selecting influential users| 0.01| RTX3090 |
> |Capturing user preferences| 1.64| A100 |
>
> As suggested, we also compare our selection of influential users with standard active learning by adopting the widely used uncertainty sampling [1]. Specifically, user uncertainty is calculated based on their average item scores, and users with the highest uncertainty are selected, with all other procedures remaining unchanged. The results presented below demonstrate that our method of selecting influential users leads to better overall performance, while maintaining comparable time costs, facilitated by PyTorch’s automatic vectorized computation.
>
> |   | R@50  | R@100 | CE@50 | CE@100 | User selection time (seconds)|
> |----|--------|-----------|-------|-------|-------|
> |Uncertainty|0.0569|0.0832|1.3219|1.3779|6.7|
> |Ours|**0.0586**|**0.0841**|**1.3275**|**1.3852**|37.6|
>
> > W3: Additional exploration of synthetic item selection.
>
> Thank you for this insightful comment. To further evaluate the effectiveness of our design in synthetic data generation, we conduct a comparison with two more complex mechanisms. First, we adopt the item scoring strategy from [2], which utilizes the Clayton copula function to account for both diversity and relevance. Second, following [3], we utilize an LLM to select items that a user might be interested in from a further filtered item candidate pool. We keep all other settings consistent with the original ToP-Rec and report the performance comparison on Twitter below. We observe that our original design maintains strong performance even when compared to these more complex methods. The main reason is that user preferences can be effectively inferred through ToP, which enhances the quality of the candidate item set and ensures reliable performance in our synthetic data generation.
>
> |   | R@50  | R@100 | CE@50 | CE@100 |
> |----|--------|-----------|-------|-------|
> |Clayton copula|0.0582|0.0838|1.2996|1.3685|
> |LLM selection |0.0576|0.0833|1.3237|1.3802|
> |Ours|**0.0586**|**0.0841**|**1.3275**|**1.3852**|
>
>
> [1] A Survey on Active Learning: State-of-the-Art, Practical Challenges and Research Directions
>
> [2] Relevance meets Diversity: A User-Centric Framework for Knowledge Exploration through Recommendations
>
> [3] LLMRec: Large Language Models with Graph Augmentation for Recommendation

---

> > ### Author Response · Authors · 2025-08-07
> >
> > Dear Reviewer,
> >
> > We appreciate your recognition of the importance of our work and the time you took to provide a detailed review. As the discussion period draws to a close, we would like to know if our previous rebuttal has addressed your concerns. If there is anything more we can do to help you in evaluating our paper, please don't hesitate to let us know.
> >
> > Best regards, Authors

---

### Official Review · Reviewer_K1gm · 2025-07-03

**Clarity:** 3
**Significance:** 3
**Originality:** 3
**Rating:** 4
**Confidence:** 4

**Summary:**

This paper introduces ToP-Rec, a novel LLM-based framework designed to enhance recommendation diversity while preserving relevance. The authors designed a Tree of Preferences, a hierarchical preference structure constructed by prompting LLMs to reason about user interactions and uncover underexplored preferences from observed behaviors. By leveraging ToP, the method identifies latent user preferences and generates synthetic interactions that augment existing data, modeling user preferences from coarse to fine. Extensive evaluations of both diversity and relevance demonstrated that the proposed approach outperforms existing methods in most cases with efficient inference latency.

**Questions:**

Aside from the questions in Weaknesses, I have the following questions:
- How does ToP-Rec perform with smaller or less capable LLMs? Qwen2.5-32B is powerful but costly. Can your method maintain reasonable performance with smaller models? How robust is ToP-Rec to LLM inaccuracies?
- Minor question out of curiosity: If a user shows interest in multiple, contradictory topics (e.g., budget travel and luxury goods), how does ToP-Rec resolve this during item selection?

**Ethical Concerns:**

["NO or VERY MINOR ethics concerns only"]

**Final Justification:**

Given the strengths and the addressed concerns, I will keep my score to indicate my recommendation of the paper.

**Limitations:**

yes

**Paper Formatting Concerns:**

I have no concern regarding the formatting.

**Quality:**

3

**Strengths And Weaknesses:**

Strengths:
- The problem of balancing diversity and relevance in recommendations is clearly stated and well-motivated. The challenges and the literature are fully discussed.
- The design of the Tree of Preferences is novel and effectively combines LLM reasoning with preference modeling.
- The methodology section is well-explained and solidly grounded.
- The strong performance on the three datasets in comparison with both heuristic methods and LLM-based diversified recommenders validates the effectiveness of the proposed method.
- The paper is well-written. It is a pleasure to read the paper.


Weaknesses:
- For the design of the relevance score, this work uses the semantic similarity of the user and item. How to ensure the quality of user and item textual attributes is a concern.
- The paper does not analyze failure cases, e.g., when diversity gains hurt relevance. Examples of incorrect LLM inferences or misaligned recommendations, if they exist, would help calibrate expectations. It would also be beneficial to show actual ToP trees and user-item matches to improve interpretability.
- The choice of the evaluation metrics could be further justified. Is category-entropy alone expressive enough to showcase the diversity of the recommendation results?

---

> ### Author Rebuttal · Authors · 2025-07-31
>
> >W1: Quality of user and item textual attributes.
>
> Thank you for your feedback. Following prior work [1], we enhance the quality of user and item textual data with the aid of LLMs. Specifically, we leverage an LLM to generate user summaries based on their profile and the textual attributes of previously interacted items, as well as to generate item summaries from their textual attributes. By utilizing the LLM, we aim to reduce noise, redundancy, and inconsistencies in the original text, leading to improved summaries that retain key semantic information. We will update the paper to include these refinement details.
>
>
> >W2: Analysis of failure cases and demonstration of ToP and user-item matches.
>
> Thank you for this insightful suggestion. To address cases where diversity gains negatively impact relevance, we conducted a post-hoc failure analysis on Twitter. Our investigation reveals that in 2.1% of cases (45/2,118 users), diversity improvements incurred relevance costs, quantified as an average recall decrease of 17.2%, accompanied by entropy increases of 14.2%. This trade-off is primarily driven by two factors: First, we find that more than half of these cases arise from the user's sparse interaction histories or the presence of uninformative interacted items, which leads to preference misinference. Second, for users with broad but shallow interaction histories, the risks of exploration boundaries may cause diversification to outweigh relevance, leading to a mismatch with their actual preferences.
>
> Due to text-only constraints, we provide a partial illustration of the actual ToP used in the Twitter dataset. We will include visualizations of the ToP structures and visual user-item matches in the manuscript to improve interpretability.
>
> ```
> ROOT
> ├── Political Systems & Governance
> │   ├── Governmental Actions and Decisions
> │   │   ├── Policy Decisions
> │   │   │   ├── Healthcare Policy
> │   │   │   │   ├── ......
> │   │   │   ├── Tax & Environment
> │   │   │    ......
> │   │   ├── Public Statements
> │   │       ├── ......
> │   ├── Political Figures
> │   │   ├── Prime Ministers and Presidents
> │   │   │   ├── .....
> │   │   ├── Cabinet Members
> │   │   ......
> │   ├── Political Events
> │   │   ├── Parties and Gatherings
> ......
> ├── Cultural Arts & Entertainment
>     ├── Movies & Television
>     │   ├── Streaming Series
>     │   ├── Theatrical Releases
>     ...
>     ├── Music
>     ....
> ```
>
> >W3: Evaluation metrics of diversity.
>
> Thank you for raising this insightful question. As a widely used metric [2,3,4], category entropy captures the distributional spread of item categories within a recommendation list, which aligns directly with the goal of diversity in our work.
>
> To provide a more comprehensive evaluation, we also implemented two additional diversity metrics: the Gini Index (GI) [5] and Intra-List Distance (ILD) [6]. The following table presents the corresponding results on Twitter, comparing Top-Rec with comparable baselines and the backbone recommender. The results demonstrate that our method consistently outperforms others, highlighting its effectiveness.
> |   | Gini@50 $\downarrow$ | Gini@100 $\downarrow$ | ILD@50 | ILD@100 |
> |----|--------|-----------|-------|-------|
> |LightGCN$^*$  |0.6763|0.6580|1.1756|1.1959|
> |LCD-UC|0.6723|0.6517|1.1849|1.2081|
> |LLM4Re-A|0.6733|0.6521|1.1775|1.1967|
> |Ours|**0.6678**|**0.6450**|**1.1954**|**1.2112**|
>
> >Q1: Performance with smaller LLMs and robustness with LLM inaccuracies.
>
> We evaluate our method with three smaller LLMs, including Qwen-2.5-7B-Intstruct, Qwen-3-8B, and Llama-3-8B-Instruct. The following table reports their performance on Twitter, compared with two comparative baselines and the backbone recommender. The results show that even with small LLMs, ToP-Rec can still outperform baselines in most cases, further demonstrating its robustness.
>
> |   | R@50  | R@100 | CE@50 | CE@100 |
> |----|--------|-----------|-------|-------|
> |LightGCN$^*$  |0.0567|0.0830|1.2841|1.3413|
> |LCD-UC|0.0517|0.0768|1.3154|1.3784|
> |LLM4Re-A|0.0562|0.0827|1.2855|1.3424|
> |Qwen3-8B|0.0578|0.0837|1.3227|1.3792|
> |Llama3-8B-Instruct|0.0573|**0.0842**|1.3089|1.3696|
> |Qwen2.5-7B-Intstruct|$\underline{0.0579}$|0.0836|**1.3291**|**1.3857**|
> |Qwen2.5-32B-Instruct (Ours)|**0.0586**|$\underline{0.0841}$|$\underline{1.3275}$|$\underline{1.3852}$|
>
> The robustness of ToP-Rec can be attributed to the design of the tree of preference, which acts as a vehicle to help the LLM systematically reason through the user's rationale and uncover their preferences, thereby mitigating the potential impact of hallucinations or inaccuracies.
>
> >Q2: Resolution for contradictory preferences.
>
> Thank you for this enlightening question. While a user's preferences may seem contradictory at first, we believe these apparent contradictions can often be rationally explained, provided the user’s choices are driven by reason rather than randomness. For example, a user interested in both budget travel and luxury goods might value practicality and exclusivity in different contexts.
>
> To explore this further, we analyzed how ToP-Rec handles such contradictions in practice. Our findings show that ToP-Rec is generally capable of rationalizing these preferences. For instance, we observed that some users may purchase budget-friendly daily items while also buying higher-end goods. ToP-Rec interprets this as a preference for affordable consumables and premium, long-term items. During item selection, ToP-Rec considers the context of different items,  seeks to provide recommendations that align with their preferences.
>
>
> [1] LLMRec: Large Language Models with Graph Augmentation for Recommendation
>
> [2] Dual-Process Graph Neural Network for Diversified Recommendation
>
> [3] Controlling Diversity at Inference: Guiding Diffusion Recommender Models with Targeted Category Preferences
>
> [4] DGCN: Diversified Recommendation with Graph Convolutional Networks
>
> [5] Post Processing Recommender Systems for Diversity
>
> [6] Avoiding monotony: improving the diversity of recommendation lists

---

> > ### Author Response · Authors · 2025-08-07
> >
> > Dear Reviewer,
> >
> > We sincerely appreciate your thoughtful review and the time invested in evaluating our work. We'd like to confirm whether our rebuttal has addressed your concerns or requires further clarification to assist your evaluation. If there is anything more we can do to help, please don't hesitate to let us know.
> >
> > Best regards, Authors

---

> > ### Comment · Reviewer_K1gm · 2025-08-07
> >
> > Thank you for the response. Most of my questions are properly answered. Since I have already given a recommendation score for the paper, I will thus keep my score.

---

> > > ### Author Response · Authors · 2025-08-09
> > >
> > > We appreciate your prompt response and the encouraging feedback on our manuscript. Your insightful comments are invaluable in enhancing our work.

---

### Decision · Program_Chairs · 2025-09-17

**Decision:**

Accept (poster)

**Comment:**

This paper introduces a novel LLM-based framework that addresses data bias in recommendation systems by constructing hierarchical "Tree of Preferences" structures to uncover users' underexplored interests and generate synthetic training data for improved diversity. The approach demonstrates strong empirical results, outperforming existing methods in 11 out of 12 comparisons across multiple datasets, while maintaining a practical separation between expensive LLM reasoning and efficient online serving. The work is technically sound with comprehensive evaluations and represents an original contribution to the intersection of LLMs and recommendation systems. While reviewers raised concerns about potential test set contamination, evaluation bias in offline datasets, and scalability constraints related to prompt length limitations, the authors provided satisfactory rebuttals that addressed most technical issues. Given the unanimous positive ratings from experienced reviewers, strong experimental validation, and the novelty of using LLMs to tackle recommendation diversity challenges, the paper is recommend for acceptance to NeurIPS. The authors should incorporate the suggested clarifications about tree construction methodology and include more detailed analysis of stability across different LLM configurations in the final version.